# Privacy-Preserving Hierarchical Fog Federated Learning (PP-HFFL) for IoT Intrusion Detection

**DOI:** 10.3390/s25237296

**Published:** 2025-11-30

**Authors:** Md Morshedul Islam, Wali Mohammad Abdullah, Baidya Nath Saha

**Affiliations:** Department of Mathematics and Information Technology, Concordia University of Edmonton, Ada Blvd NW, Edmonton, AB T5B 4E4, Canada; wali.abdullah@concordia.ab.ca (W.M.A.); baidya.saha@concordia.ab.ca (B.N.S.)

**Keywords:** internet of things (IoT), intrusion detection system (IDS), fog computing, federated learning (FL), personalized federated learning (PFL), scalable IoT systems, differential privacy (DP), privacy-preserving hierarchical fog federated learning (PP-HFFL)

## Abstract

The rapid expansion of the Internet of Things (IoT) across critical sectors such as healthcare, energy, cybersecurity, smart cities, and finance has increased its exposure to cyberattacks. Conventional centralized machine learning-based Intrusion Detection Systems (IDS) face limitations, including data privacy risks, legal restrictions on cross-border data transfers, and high communication overhead. To overcome these challenges, we propose Privacy-Preserving Hierarchical Fog Federated Learning (PP-HFFL) for IoT intrusion detection, where fog nodes serve as intermediaries between IoT devices and the cloud, collecting and preprocessing local data, thus training models on behalf of IoT clusters. The framework incorporates a Personalized Federated Learning (PFL) to handle heterogeneous, non-independent, and identically distributed (non-IID) data and leverages differential privacy (DP) to protect sensitive information. Experiments on RT-IoT 2022 and CIC-IoT 2023 datasets demonstrate that PP-HFFL achieves detection accuracy comparable to centralized systems, reduces communication overhead, preserves privacy, and adapts effectively across non-IID data. This hierarchical approach provides a practical and secure solution for next-generation IoT intrusion detection.

## 1. Introduction

The Internet of Things (IoT) represents a vast ecosystem of interconnected smart devices that autonomously collect, share, and process data over the internet without direct human intervention [1]. This IoT paradigm has transformed numerous sectors, including healthcare, energy, transportation, smart cities, and finance, enabling real-time monitoring, automation, and data-driven decision-making. However, widespread adoption of IoT also expands the attack surface, exposing devices and networks to diverse cyber and privacy threats [2]. In healthcare IoT, for instance, sensors embedded in wearable devices or medical equipment can reveal highly sensitive information such as health conditions, daily routines, and precise locations. Notable incidents, such as the 2021 Verkada breach [3], compromised live feeds from 150 million surveillance cameras, highlighting the urgency for robust security and privacy-preserving mechanisms. Developing secure, privacy-aware IoT infrastructures is therefore critical to ensuring public trust and enabling its large-scale adoption.

To counter cyberattacks in IoT ecosystems, Intrusion Detection Systems (IDS) have become a primary defense mechanism, particularly those based on anomaly detection [4,5]. Such systems monitor behavioral patterns and flag deviations as potential intrusions, often combining anomaly-based methods with signature-based techniques to improve accuracy. Traditional IDS architectures predominantly rely on centralized machine learning (ML), where IoT devices transmit raw data to a cloud server for training and inference. This approach suffers from several limitations: (i) sensitive raw data is exposed to third-party servers, creating privacy risks; (ii) compliance with cross-border data protection regulations becomes challenging; (iii) high network and computation overhead arises from transmitting large and heterogeneous IoT datasets. Decentralized approaches, which perform processing closer to the data source, such as at the edge or fog layer, offer a promising alternative to mitigate these issues.

Federated Learning (FL) [6] addresses these limitations by enabling multiple clients to collaboratively train a global model without sharing raw data. In FL, a central server initializes a global IDS model and distributes it to participating clients. Each client performs local training on private datasets and sends only model updates (weights or gradients) back to the server. Aggregation methods, such as Federated Averaging (FedAvg) [7], combine these updates to refine the global model iteratively. This process continues until the model converges or reaches desired accuracy, improving its generalization to unseen attack patterns.

Several FL-based IDS frameworks for IoT have been developed [8,9,10,11,12]. Nevertheless, these methods often rely on resource-constrained IoT devices as FL clients, which presents challenges such as limited computation, energy restrictions, and non-independent and identically distributed (non-IID) settings. To address these limitations, we propose a Privacy-Preserving Hierarchical Fog Federated Learning (PP-HFFL) framework for IoT intrusion detection, which integrates the advantages of Fog-based Federated Learning (Fog-FL) while incorporating privacy-preserving mechanisms. In PP-HFFL, fog nodes-positioned between IoT devices and the cloud-function as local aggregators and decision-makers, while IoT devices primarily collect and preprocess data. This hierarchical approach, PP-HFFL, reduces the computation overhead of IoT devices by offloading computation to fog nodes, and enhances scalability. Additionally, fog nodes enable near-real-time intrusion detection by processing data close to the source and responding rapidly to anomalous events.

Despite the benefits of PP-HFFL, several challenges persist, particularly at the fog layer. The most critical include non-IID data [13], system scalability, and data privacy [14]. Addressing these issues is essential to ensuring PP-HFFL–based IDS frameworks remain effective, secure, and reliable across heterogeneous IoT environments. This study systematically investigates these challenges and proposes solutions validated on real-world IoT benchmark datasets.

**Non-IID Data:** In PP-HFFL, each fog node aggregates updates from multiple heterogeneous IoT devices, often resulting in skewed or imbalanced class distributions and, in extreme cases, missing classes on certain clients. Such non-IID conditions can lead to biased model updates, slower convergence, and reduced global model accuracy. The severity of these effects depends on the complexity of the dataset and the degree of distributional heterogeneity, motivating a detailed analysis of non-IID impacts in hierarchical Fog-FL systems.**Scalability:** Scalability in PP-HFFL requires accommodating a variable number of fog nodes and managing dynamic node participation. Increasing the number of fog nodes can improve learning capacity, as observed similarly in other scalable systems [15], but it may also intensify data and model heterogeneity. Additionally, nodes may join or leave during training, necessitating robust coordination mechanisms to preserve stable convergence. Therefore, evaluating scalability under diverse participation patterns is essential for practical and reliable deployment.**Data Privacy:** Although FL reduces privacy risks by keeping data local, interactions at the fog layer can introduce new attack surfaces. Malicious fog nodes could infer sensitive patterns from model updates or manipulate aggregation results. PP-HFFL integrates DP mechanism to maintain strong data privacy guarantees without significantly compromising model utility.

The primary objective of this research is to design and evaluate the PP-HFFL–based IDS that addresses the identified challenges. Specifically, we (i) examine the effect of non-IID data on detection accuracy, (ii) evaluate system scalability under varying fog node participation, and (iii) incorporate DP at the fog layer to ensure data privacy. Experiments on two IoT benchmark datasets demonstrate the strong performance and practical applicability of the proposed PP-HFFL approach.

The remainder of this paper is structured as follows: Section 2 presents background concepts, Section 3 surveys related works, Section 4 details the architecture of the proposed PP-HFFL framework, Section 5 reports experimental evaluations and discussions, and Section 6 concludes with key findings and future directions.

## 2. Background

### 2.1. Federated Learning

Federated Learning (FL), first introduced by Google in 2017, was designed to allow Android users to collaboratively train models without sharing personal data [7]. FL represents a privacy-preserving, decentralized paradigm of ML, where a global model is trained across multiple clients without transferring raw data to a central server. Each client independently optimizes a local objective function-typically through stochastic gradient descent (SGD)-and sends the resulting model updates or gradients to a coordinating server. The server then performs an aggregation step, usually by averaging the received parameters, to update the global model. This process is repeated iteratively until convergence.

Under appropriate data distributions and system configurations, federated learning can achieve detection performance comparable to centralized machine learning. At the same time, FL inherently provides stronger privacy guarantees, since raw data remain local to each device. In contrast, centralized ML typically requires additional privacy-preserving mechanisms [16] before data can be transmitted to a central server. The foundational FL formulation proposed by [7] is defined as(1)$$f\left(w\right)=\sum_{k=1}^{K}\frac{n_{k}}{n}F_{k}\left(w\right),\;\mathrm{where}\;F_{k}\left(w\right)=\frac{1}{n_{k}}\sum_{i\in D_{k}}f_{i}\left(w\right),$$
where $f_{i}\left(w\right)$ denotes the loss function for the *i*-th training sample $\left(x_{i},y_{i}\right)$ parameterized by w. Here, *K* represents the total number of participating clients, and $F_{k}\left(w\right)$ is the local objective of client *k*, which contains $n_{k}$ samples stored in its dataset $D_{k}$. The total dataset size across all clients is $n=\sum_{k=1}^{K}n_{k}$.

A common aggregation rule in FL is the Federated Averaging (FedAvg) algorithm [7], where the global model update after each communication round *t* is computed as(2)$$w_{t+1}=\sum_{k=1}^{K}\frac{n_{k}}{n}w_{t}^{\left(k\right)},$$
where $w_{t}^{\left(k\right)}$ is the locally updated parameter vector of the *k*-th client after completing its local training in round *t*. This weighted averaging ensures that clients with larger datasets contribute proportionally more to the global update.

Despite its advantages, FL faces several challenges, including client heterogeneity, and data non-IID characteristics, especially in IoT environments, where clients are resource-constrained devices with intermittent connectivity [14,17].

### 2.2. Non-IID Properties of IoT Data

In a Fog-FL-based IDS, fog nodes aggregate data from nearby IoT devices and act as clients in the federated setup. However, some fog clients may only receive data samples belonging to a limited subset of malware classes, leading to a label imbalance across clients. Moreover, certain types of attacks appear more frequently in practice, creating an overall class imbalance at the system level. When both types of imbalance coexist, the heterogeneity-or non-IID nature-of the data is further exacerbated [18,19].

To model such non-IID distributions in FL simulations, one of the widely adopted strategies is uniform label assignment. Label assignment refers to the way class labels are distributed across clients in FL, which directly influences the degree of data heterogeneity. In our case, by adopting a uniform label assignment strategy, we created a controlled non-IID setting. Beyond label imbalance, non-IID data may also arise due to the following:Dirichlet sampling: Stochastically partitions data according to a Dirichlet (α) distribution.Covariate shift: The input feature distribution P(x) varies across clients while the conditional label distribution P(y|x) remains constant.Concept shift: Clients share the same feature distribution but differ in label assignments, i.e., P(y|x) changes across clients.

These advanced distribution shifts introduce significant challenges in achieving convergence and fairness in global optimization, but are beyond the immediate scope of this work.

### 2.3. Personalization in Federated Learning

Traditional FL seeks to learn a single global model $w_{g}$ that performs well across all clients. However, in the presence of highly heterogeneous data, a single model often underperforms for certain clients, as it cannot fully capture their local data characteristics. Conversely, training individual local models $w_{k}$ in isolation may lead to overfitting and poor generalization.

To balance this trade-off, Personalized Federated Learning (PFL) [20,21] introduces adaptation mechanisms that tailor the global model to the data distribution of each client. The general objective of PFL can be represented as(3)$$\min_{\left\{w_{k}\right\},w_{g}}\sum_{k=1}^{K}\frac{n_{k}}{n}\left(F_{k}\left(w_{k}\right)+\lambda {∥w_{k}-w_{g}∥}^{2}\right),$$
where λ is a regularization parameter that controls the closeness between the local and global models. Smaller λ encourages greater personalization, while larger values enforce a stronger alignment with the global model.

There are numerous personalization approaches proposed in the literature, such as client clustering, local fine-tuning, model interpolation, and meta-learning-based adaptation. Among these, one effective strategy involves client clustering [22,23], which groups clients with similar data distributions or geographical proximity. In a Fog-FL-based IoT system, such clustering occurs naturally: IoT devices within the same fog domain often share environmental and traffic characteristics. By fine-tuning the pre-trained fog model on its local data, each fog node can achieve both personalization and scalability in intrusion detection tasks.

### 2.4. Differential Privacy

Differential privacy (DP) is a rigorous mathematical framework designed to protect individual data contributions while allowing useful aggregate analysis [24,25]. A mechanism M satisfies (ϵ,δ)-DP if, for all neighboring datasets *D* and $D^{\prime }$ differ by one record, and for all output subsets *S*:(4)$$\mathrm{Pr}\left[M\left(D\right)\in S\right]\leq e^{\epsilon }\mathrm{Pr}\left[M\left(D^{\prime }\right)\in S\right]+\delta ,$$
where ϵ controls the privacy–utility trade-off (smaller ϵ implies stronger privacy), and δ represents a small probability of failure.

In the central differential privacy (CDP) setting, a trusted server collects raw data and adds noise to the output before release. In contrast, local differential privacy (LDP) ensures that each client perturbs its own data or gradients before sharing, offering privacy even from the server [26].

In Fog-FL, both models are applicable. A fog node can apply CDP when aggregating data from IoT devices or LDP when acting as a client that perturbs its own gradient updates. Standard mechanisms for adding DP noise include the Laplace and Gaussian mechanisms [27], where random noise proportional to the query’s sensitivity is injected.

To manage privacy loss across multiple training rounds, Rényi differential privacy (RDP) [28] provides a tighter composition bound. The (α,ϵ)-RDP guarantees for a mechanism M is defined as(5)$$D_{\alpha }\left(M\left(D\right)∥M\left(D^{\prime }\right)\right)\leq \epsilon ,$$
where $D_{\alpha }\left(\cdot ∥\cdot \right)$ denotes the Rényi divergence of order α. RDP is particularly suitable for DP-SGD implementations (e.g., TensorFlow Privacy, Opacus) used in FL, as it efficiently tracks cumulative privacy loss across communication rounds.

In summary, integrating RDP-based noise mechanisms within Fog-FL architectures strikes a balance between privacy protection, model utility, and computational feasibility, making it suitable for IoT-based intrusion detection systems.

## 3. Related Work

Federated Learning (FL) has recently gained attention as an effective approach for enabling collaborative intrusion detection in distributed and privacy-sensitive IoT and industrial IoT environments. Unlike traditional centralized training, FL allows each device to learn from local data and exchange only model updates with a coordinating server, thereby preserving data confidentiality while still benefiting from shared intelligence across the network [29].

Several studies have explored FL-based IDS designs for diverse IoT scenarios. For instance, FLEAM [30] was developed for IoT-based Distributed Denial of Service (DDoS) attack detection by integrating FL with edge analytics to counter large-scale attacks efficiently. Federated mimic learning [31] introduced a teacher–student knowledge distillation mechanism to improve IDS accuracy while maintaining data confidentiality. Similarly, DeepFed [32] applied deep learning-based FL within industrial cyber–physical systems, demonstrating the potential of collaborative anomaly detection across heterogeneous industrial sites. In the agricultural domain, FELIDS [10] extended FL to smart farming scenarios, showcasing lightweight intrusion detection for resource-constrained devices and emphasizing scalability in distributed environments. Furthermore, a privacy-preserving FL-based IDS was proposed in [33] to secure IoT systems without exposing sensitive local data. More recently, FL-IDS [34] further advanced this line of research by incorporating FL into IoT-based IDS with a focus on maintaining high detection accuracy across heterogeneous client devices and non-uniform environments.

Despite these advances, most conventional FL-based IDS still face critical challenges, including synchronization delays, non-IID data, and constrained edge resources. IoT devices inherently exhibit limitations in computing, memory, energy, bandwidth, and hardware diversity, compounded by statistical heterogeneity in local data distributions. These constraints make standard FL protocols difficult to deploy effectively at the edge, as emphasized in [35]. Recent studies address limited device capacity through model-level optimizations for edge deployment. Resilience-focused methods adapt model complexity to stabilize inference under varying device conditions [36], and lightweight inference or compression techniques reduce computation and energy demands [37]. All these efforts, however, emphasize general robustness rather than collaborative intrusion detection across distributed IoT nodes, and thus complement rather than replace FL-based IDS research. Likewise, endogenous security approaches embed adaptive, immune-inspired defenses into industrial IoT systems [38], often incorporating FL with blockchain trust models, but their core aim is system-level self-protection rather than intrusion detection, and is therefore outside the scope of our FL-based IDS design.

### 3.1. Fog-Enabled Federated Learning for IoT IDS

To mitigate resource constraints, as well as scalability and privacy issues, in IoT-based IDS, several studies have proposed integrating fog computing with federated learning. The fog layer, positioned between the edge and the cloud, provides intermediate computation, storage and coordination capabilities-making it particularly suitable for latency-sensitive IoT security applications.

Javeed et al. [39] incorporated fog computing into FL to offload intensive training tasks from resource-limited IoT devices, thereby reducing latency and improving overall IDS performance. Bensaid et al. [40] extended this concept by securing IoT systems through fog-layer FL deployment, achieving collaborative intrusion detection while preserving client privacy. Similarly, Liu et al. [41] investigated fog-client selection strategies-both random and resource-aware-demonstrating how optimized fog participation can improve detection efficiency. A hierarchical federated structure, Fog-FL [42], was proposed to further improve scalability, where geographically distributed fog nodes perform local aggregation and synchronization with the cloud. This design effectively aligns FL with edge constraints. In a similar vein, de Souza et al. [43] developed a fog-based FL framework for IDS that exploits fog-layer processing to enhance scalability, responsiveness, and distributed model accuracy in large-scale IoT deployments.

In addition to these approaches, Abdel-Mageed et al. [44] proposed a privacy-preserving fog–federated IDS that jointly addresses non-IID data distribution and adversarial data leakage through the integration of generative adversarial networks (GANs) and differential privacy. Their work highlights the growing emphasis on designing hybrid privacy mechanisms at the fog layer to balance learning efficiency with confidentiality, particularly in heterogeneous and dynamic IoT ecosystems.

Table 1 summarizes representative fog-enabled FL approaches for IDS. These systems collectively illustrate how bringing computation closer to data sources can alleviate the bottlenecks of traditional FL. By minimizing latency, and adapting to edge-level heterogeneity, fog-based FL significantly enhances responsiveness and scalability for IoT intrusion detection. Nonetheless, as summarized in the table, several limitations persist: most studies provide limited treatment of non-IID data handling, only partially address scalability in dynamic network topologies, and often overlook end-to-end privacy preservation at the fog layer. Compared to the broader FL-IDS literature, relatively few studies explicitly focus on deployment at the fog layer. This gap presents a significant opportunity for future research to develop more robust, privacy-preserving, and adaptive fog federated IDS frameworks—for example, by incorporating techniques such as differential privacy [45], dimension reduction [16], and random-projection-based noise addition [46] directly at the fog layer.

### 3.2. Summary and Research Gap

In summary, the existing literature has established the potential of FL and fog computing as enablers of collaborative and privacy-aware intrusion detection in IoT ecosystems. However, the reviewed studies reveal that most frameworks prioritize performance, with only a few addressing scalability and/or privacy; rarely are all three properties achieved simultaneously under realistic non-IID conditions. Furthermore, current fog federated systems often lack adaptive mechanisms to dynamically balance convergence speed. These challenges underscore the need for a unified architecture that jointly optimizes non-IID robustness, scalability, and privacy preservation.

Addressing these issues forms the central motivation of this work, which proposes a Privacy-Preserving Hierarchical Fog Federated Learning (PP-HFFL) framework. PP-HFFL is designed to provide scalable, privacy-preserving, and adaptive intrusion detection in large-scale IoT networks, capable of operating efficiently under heterogeneous, adversarial, and real-world deployment scenarios.

## 4. PP-HFFL: Privacy-Preserving Hierarchical Fog Federated Learning for IDS

Building upon insights and limitations identified in the existing literature, this section introduces the proposed PP-HFFL framework for IDS. The methodology is designed to explicitly address challenges in scalability, heterogeneous data, and privacy leakage, combining the hierarchical advantages of fog computing with the collaborative intelligence of federated learning. PP-HFFL enhances efficiency, accuracy, and adaptability while maintaining privacy guarantees. The subsections below describe the system architecture, underlying algorithms, implementation strategies, and security and privacy analyses.
*System Assumptions.* Several key assumptions regarding the system entities, operational environment, and trust model are outlined:
Data Assumptions: IoT-collected datasets represent diverse behavioral and operational patterns, which are privacy-sensitive. Aggregated datasets at the fog level are inherently non-IID due to (i) each fog node observing distinct subsets of attack and benign classes, and (ii) imbalanced class distributions both across and within clients. Data drift may occur over time as device behaviors evolve or new IoT devices join the network.Trust Assumptions: Each fog client is trusted by its associated IoT devices. All other fog nodes and the central cloud server are semi-honest (honest-but-curious), meaning they follow the protocol but may attempt to infer privacy-sensitive information from updates. No entity is fully malicious or colluding unless explicitly defined in the threat model.Computation Assumptions: IoT edge devices are resource-constrained, with limited processing power, memory, and energy, and cannot efficiently train complex ML models. Fog nodes have moderate computational resources to perform local training and communication with both IoT devices and cloud server, while the cloud server has sufficient computational capacity for global coordination and aggregation.Communication Assumptions: IoT-to-fog communication is bandwidth-limited and may experience latency or data loss. Fog-to-cloud links are relatively stable, leveraging high-speed backhaul. Synchronization between fog and cloud layers is periodic, conserving bandwidth while enabling efficient model updates.Security and Privacy Assumptions: Standard cryptographic mechanisms (e.g., secure aggregation) are assumed to protect local updates at the server side and prevent model inversion attacks. Securing key exchange and authentication exist between fog nodes and cloud prevents impersonation or poisoning attacks.Deployment Assumptions: Each fog node serves a fixed set of IoT clusters. The number of fog nodes may scale dynamically based on network density. Time synchronization across nodes is loosely coordinated to allow asynchronous or semi-synchronous federated updates.

### 4.1. System Architecture of PP-HFFL

PP-HFFL leverages fog-enabled federated learning for large-scale IoT intrusion detection. Unlike conventional cloud-only systems, fog nodes are positioned closer to edge devices, with moderate computational capabilities for localized data processing and training. This reduces dependence on centralized resources, enabling near-real-time intrusion detection. Figure 1 illustrates the hierarchical three-tier architecture: Cloud Tier, Fog Tier, and Edge Tier, where each layer has distinct roles to ensure scalable, privacy-preserving, and accurate federated learning under non-IID conditions.

*Cloud Tier in PP-HFFL.* The cloud tier serves as the central coordinator and aggregator. It initializes the global IDS model, optionally pre-trains it, and distributes it to participating fog nodes. During each global round, it collects updated model parameters from fog nodes, aggregates them using FedAvg algorithm, updates the global model, and redistributes it. When new fog nodes join, the cloud provides them with the most recent global model, ensuring smooth onboarding and maintaining scalable, continuous federated training.

*Fog Tier in PP-HFFL.* The fog tier consists of geographically distributed fog nodes that act as intermediate computation layers between IoT devices and the cloud. Each fog node has the following functions:Receives the global model from the cloud.Performs local training with its own dataset.Applies DP via Gaussian noise to gradients before sending them to the cloud.Optionally personalizes the model to adapt to local non-IID data distributions.This process repeats iteratively across multiple communication rounds until convergence. New fog nodes seamlessly integrate into the PP-HFFL network.

*Edge Tier in PP-HFFL.* IoT devices continuously collect raw data, perform lightweight preprocessing (feature extraction, normalization, noise filtering), and transmit processed data to their assigned fog node. Edge devices do not participate directly in federated training due to resource constraints.

### 4.2. Federated Learning Algorithm in PP-HFFL

The training process of the PP-HFFL is governed by the Federated Averaging with differential privacy algorithm (FedAvgDP), described in Algorithm 1. Key steps include the following: (i) Cloud selects a subset of fog clients for each round. (ii) PrivacyEngine adds Gaussian noise to gradients controlled by ϵ and δ. (iii) Local training for *E* epochs on mini-batches. (iv) Local gradients are clipped, noised, and used to update local models. (v) Fog nodes send updated models to the cloud, which aggregates them using FedAvg. (vi) The updated global model’s parameters is broadcasted to all fog nodes. (vii) Optional personalization at fog nodes to handle non-IID data.

*Privacy, Scalability, and Adaptability in PP-HFFL.* PP-HFFL addresses non-IID data, ensures scalable and adaptive operation, and preserves privacy. DP secures model updates, dynamic node addition enables scalability, and personalization improves convergence and accuracy. Fog-layer computation reduces network traffic and latency, supporting real-time intrusion detection.
**Algorithm 1** PP-HFFL: Federated Averaging with Differential Privacy (*FedAvgDP*) at Fog Nodes**Require:** *C*: set of fog clients, *G*: global dataset, *M*: global model with parameters $w^{\left(0\right)}$, *R*: max rounds, η: learning rate, γ: client ratio, *E*: local epochs, ϵ: privacy parameter, δ: failure probability parameter.**Ensure:** Differentially private, personalized updated global model *M*  1:**for** r=1 to *R* **do**  2:    $C_{r}\leftarrow \mathrm{RandomSample}\left(C,\gamma |C|\right)$▹ Select subset of fog clients  3:    L←{}▹ Initialize list for local models  4:    **for** each fog client $c\in C_{r}$ **do**  5:        Initialize $M_{c}\leftarrow M$▹ Client receives global model from cloud  6:        Let $w_{c}$ be the parameters of $M_{c}$  7:        Attach PrivacyEngine (ϵ,δ)▹ Enable DP at client side  8:        **for** t=1 to *E* **do**  9:           Sample batch (x,y) from local dataset $D_{c}$10:           Compute predictions $\hat{y}\leftarrow M_{c}\left(x\right)$11:           Compute loss $\ell \leftarrow L(\hat{y},y)$12:           Compute per-sample gradients $g_{i}\leftarrow \nabla \ell_{i}$13:           Clip gradients: $g_{i}\leftarrow g_{i}/\max(1,∥g_{i}{∥}_{2}/C_{\max})$14:           Add Gaussian noise: $\tilde{g}=\frac{1}{\left|B\right|}\sum_{i\in B}g_{i}+N\left(0,\sigma^{2}C_{\max}^{2}I\right)$15:           Update local weights: $w_{c}\leftarrow w_{c}-\eta \cdot \tilde{g}$16:        **end for**17:        Apply personalization layer to client *c*’s model▹ adapted to local data as in Section 4.2.118:        Evaluate local model $M_{c}$ and record metrics19:        $L\left[c\right]\leftarrow w_{c}$20:    **end for**21:    **Aggregate:** $w^{\left(r\right)}\leftarrow \frac{1}{|C_{r}|}\sum_{c\in C_{r}}L\left[c\right]$▹ FedAvg aggregation at fog layer22:    Update *M* with aggregated parameters $w^{\left(r\right)}$ and evaluate on global dataset *G*23:**end for**24:**return** *M*▹ Differentially-private, personalized global model

#### 4.2.1. Mathematical Formulation of the Personalization Algorithm

*Model and Objective.* Each client *k* holds local data $D_{k}={\left\{\left(x_{i}^{\left(k\right)},y_{i}^{\left(k\right)}\right)\right\}}_{i=1}^{n_{k}}$, and shares a global model $f_{\theta_{g}}:R^{d}\to R^{C}$ with parameters $\theta_{g}$ trained collaboratively in the federated setting. During the personalization stage, each client adapts $\theta_{g}$ to its local data by learning personalized parameters $\theta_{k}$, minimizing the following regularized loss:(6)$$L_{k}^{\mathrm{PFL}}\left(\theta_{k};\theta_{g}\right)=\frac{1}{n_{k}}\sum_{i=1}^{n_{k}}\ell \left(f_{\theta_{k}}\left(x_{i}^{\left(k\right)}\right),\;y_{i}^{\left(k\right)}\right)+\lambda {∥\theta_{k}-\theta_{g}∥}^{2},$$
where ℓ(·,·) denotes the cross-entropy loss and λ controls the trade-off between personalization and global consistency. A smaller λ emphasizes local adaptation, while a larger λ enforces stronger alignment with the global model. The cross-entropy loss is given by$$\ell \left(\hat{y},y\right)=-\sum_{c=1}^{C}1_{\left\{y=c\right\}}\log{\hat{y}}_{c}.$$

The corresponding gradient descent update rule is$$\theta_{k}^{\left(t+1\right)}=\theta_{k}^{\left(t\right)}-\eta \nabla_{\theta_{k}^{\left(t\right)}}\left(L_{k}\left(\theta_{k}^{\left(t\right)}\right)+\lambda {∥\theta_{k}^{\left(t\right)}-\theta_{g}∥}^{2}\right),$$
where η denotes the local learning rate.

*Accuracy Evaluation.* Model accuracy for each client *k* before and after personalization is computed as$$\begin{array}{cc} {\mathrm{Acc}}_{k,\mathrm{before}} & =\frac{1}{m_{k}}\sum_{i=1}^{m_{k}}1\left(\arg\max f_{\theta_{g}}\left(x_{i}^{\left(k\right)}\right)=y_{i}^{\left(k\right)}\right),\\ {\mathrm{Acc}}_{i,\mathrm{after}} & =\frac{1}{m_{k}}\sum_{i=1}^{m_{k}}1\left(\arg\max f_{\theta_{k}}\left(x_{i}^{\left(k\right)}\right)=y_{i}^{\left(k\right)}\right), \end{array}$$
where $m_{k}$ is the number of local test samples of user *k*.

The global summary metrics are$$\begin{array}{cc} {\bar{\mathrm{Acc}}}_{\mathrm{before}} & =\frac{1}{K}\sum_{k=1}^{K}{\mathrm{Acc}}_{k,\mathrm{before}},\\ {\bar{\mathrm{Acc}}}_{\mathrm{after}} & =\frac{1}{K}\sum_{k=1}^{K}{\mathrm{Acc}}_{i,\mathrm{after}}. \end{array}$$

*Algorithmic Description.* Personalization federated learning is described by Algorithm 2.
**Algorithm 2** Personalized Federated Learning (PFL) with Regularization**Require:** Global model $\theta_{g}$, regularization weight λ, learning rate η, epochs *E*, client set C={1,…,K}**Ensure:** Personalized models ${\left\{\theta_{k}\right\}}_{k=1}^{K}$, client accuracy metrics1:**for** each client k∈C **do**2:    Load $\left(D_{k}^{\mathrm{train}},D_{k}^{\mathrm{test}}\right)$3:    Initialize local model $\theta_{k}\leftarrow \theta_{g}$4:    Evaluate ${\mathrm{Acc}}_{k,\mathrm{before}}\leftarrow \mathrm{Evaluate}\left(\theta_{k},D_{k}^{\mathrm{test}}\right)$5:    **for** epoch =1 to *E* **do**6:        **for** each mini-batch $\left(x,y\right)\in D_{k}^{\mathrm{train}}$ **do**7:           Compute prediction $\hat{y}=f_{\theta_{k}}\left(x\right)$8:           Compute empirical loss $L_{k}=\ell \left(\hat{y},y\right)$9:           Compute regularization loss$$L_{\mathrm{reg}}=\lambda {∥\theta_{k}-\theta_{g}∥}^{2}$$10:           Total loss: $L_{k}^{\mathrm{PFL}}=L_{k}+L_{\mathrm{reg}}$11:           Update parameters:$$\theta_{k}\leftarrow \theta_{k}-\eta \nabla_{\theta_{k}}L_{k}^{\mathrm{PFL}}$$12:        **end for**13:    **end for**14:    Evaluate ${\mathrm{Acc}}_{k,\mathrm{after}}\leftarrow \mathrm{Evaluate}\left(\theta_{k},D_{k}^{\mathrm{test}}\right)$15:    Store metrics $\mathrm{metrics}\left[k\right]\leftarrow \left({\mathrm{Acc}}_{k,\mathrm{before}},{\mathrm{Acc}}_{k,\mathrm{after}}\right)$16:**end for**17:Compute global averages:$${\bar{\mathrm{Acc}}}_{\mathrm{before}}=\frac{1}{K}\sum_{k}{\mathrm{Acc}}_{k,\mathrm{before}},\;{\bar{\mathrm{Acc}}}_{\mathrm{after}}=\frac{1}{K}\sum_{k}{\mathrm{Acc}}_{k,\mathrm{after}}$$18:**return**$\left\{\theta_{k}\right\}$, metrics, and averaged accuracies

*Integration into the Fog-FL Hierarchy.* In the proposed Fog-FL framework, each fog node serves as a local aggregator for a subset of geographically or statistically similar IoT clients. After global aggregation, the personalized fine-tuning step (Algorithm 2) is executed at each fog node or end-device to adapt the global model $\theta_{g}$ to its local environment. This hierarchical adaptation allows each fog domain to retain global knowledge while optimizing for domain-specific data distributions, thereby improving accuracy and robustness under non-IID conditions typical of large-scale IoT deployments.

### 4.3. Security and Privacy Analysis in PP-HFFL

FL-based IDS in fog environments face security and privacy challenges due to decentralized, non-IID data, and multi-tier communication. PP-HFFL ensures robust intrusion detection while preserving privacy and maintaining system scalability. Differential privacy (DP) and personalized federated learning (PFL) strategies enhance model resilience and convergence under heterogeneous conditions. The hierarchical architecture enables real-time decision-making at the fog layer, reducing computation at IoT devices, and minimizing data transmission to the cloud.

*Threat Landscape.* Attacks on FL-based systems include manipulation attacks [48,49] and inference attacks [50,51]. Manipulation attacks compromise model integrity, while inference attacks attempt to extract sensitive information. Both the training and inference phases are vulnerable, necessitating multi-layered defense mechanisms [52,53].

*Mitigation via Differential Privacy.* DP noise added at the fog nodes to obscures sensitive gradient information, thereby reducing the risk of gradient-based inference attacks. Any resulting accuracy trade-offs can be managed through adaptive noise calibration or hybrid privacy schemes [44,45,54]. While homomorphic encryption can also secure model updates [55], such methods introduce substantial computational overhead and are therefore not incorporated into our current design. Rather than implementing full adversarial testing or a detailed threat model, we focus specifically on differential privacy as a lightweight mechanism for mitigating gradient leakage risks in hierarchical FL. Attacks that fall outside the protection offered by DP-such as stronger active adversaries or cryptographic-level threats-are beyond the scope of this work.

*Robustness through Personalized Federated Learning.* Personalized FL (PFL) fine-tunes the global model to each fog node’s local data and the regularization parameter. This limits the impact of poisoned updates, accelerates convergence under non-IID data, and provides implicit anomaly detection [56,57].

*Privacy Preservation and System Integrity.* Only obfuscated gradients or parameters are shared between fog and cloud layers, preventing reconstruction of raw data. PP-HFFL’s distributed design inherently limits large-scale data leakage. Compromised IoT devices can still be detected through collaboratively trained models, ensuring system privacy and integrity across the fog–cloud continuum.

## 5. Experimental Results

### 5.1. Dataset

We evaluated the proposed PP-HFEL method on the RT-IoT 2022 [4] and CIC-IoT 2023 [58] datasets, both specifically designed for IoT intrusion detection. RT-IoT 2022 is relatively smaller in sample size but richer in feature space, whereas CIC-IoT 2023 is significantly larger but with fewer features. Specifically, RT-IoT 2022 contains 79 features and 123,117 samples, while CIC-IoT 2023 has 46 features and 1,048,575 samples. Both datasets are tabular (non-image), making them less complex in raw input dimensionality compared to image corpora, yet still challenging due to heterogeneity and non-IID distributions across devices.

Both datasets exhibit strong class imbalance, which poses significant challenges for federated learning. Table 2 summarizes the sample counts per class. In CIC-IoT, several classes (e.g., DDoSICMPFlood, DDoSUDPFlood, DDoSTCPFlood) have more than 100,000 samples, whereas others (e.g., ReconPingSweep, BackdoorMalware) contain fewer than 100. Similarly, in RT-IoT, the largest class (DOSSYNHping) exceeds 94,000 samples, while smaller classes such as MetasploitBFSSH and NMAPFINScan have under 100 samples. CIC-IoT also spans a much wider label space, with 34 distinct classes compared to 12 in RT-IoT.

### 5.2. Experiment Setup

For the PP-HFFL IDS experiments, we preprocessed the datasets, designed deep neural network (DNN) architectures for both the global and local models, and carefully selected hyperparameters. Training data was partitioned across multiple fog clients, enabling hierarchical collaborative learning under various non-IID conditions. To ensure privacy, DP noise was integrated into the PP-HFFL framework, resulting in a privacy-preserving system. Additionally, personalized federated learning techniques were incorporated to improve local model performance while maintaining overall system scalability.

During federated training, each selected client executed five local epochs before uploading its differentially private model updates to the central server. The server aggregated these updates using FedAvg and broadcasted the updated global model back to all clients. This process was repeated for 300 communication rounds. The same configuration was applied across all experiments to ensure comparability.

All experiments were conducted in two distinct environments to evaluate reproducibility and to confirm that the code executes consistently across CPU and GPU-enabled setups.

Local machine: Visual Studio Code (version 1.106.2) on Windows using Windows Subsystem for Linux (WSL), Intel Core i5-1235U CPU, and 16 GB RAM. All runs used a single CPU core without multithreading. This environment served as the primary platform for developing and validating the pipeline.Cloud environment: GPU-enabled runtime provided by the Digital Research Alliance, configured with 1 GPU and 8 GB RAM. The GPU (2 GB VRAM) was used only to verify that the workflow also runs on GPU-capable hardware; no GPU-specific optimization was applied.

All hyperparameters, including learning rate, batch size, and DP noise scale, were kept constant across environments to ensure that performance differences reflected computational characteristics rather than configuration inconsistencies.

#### 5.2.1. Model Architecture and Hyperparameters

We designed a custom multi-layer perceptron (MLP) optimized for one-dimensional IoT feature vectors. The input dimension *d* is passed through a series of fully connected layers, each followed by batch normalization and ReLU activations. The final hidden layer has 128 units before the output layer, which produces logits for *n* classes. This architecture is applied consistently across global and local models, with minor adjustments for dataset-specific feature dimensions or class counts. A complete overview of the architecture and hyperparameters is provided in Table 3 and Table 4.

#### 5.2.2. Data Preprocessing

For both centralized ML models and PP-HFFL IDS experiments, we used the original RT-IoT 2022 and CIC-IoT 2023 datasets with a uniform label assignment (label split) strategy for federated learning. Centralized ML achieved 99.49% and 90.64% accuracy on RT-IoT and CIC-IoT, respectively, while FL reached above 99.0% on RT-IoT and above 85.0% on CIC-IoT under the same 300-round training configuration. The primary challenge for FL and PP-HFFL lies in the computational cost and time required for global training, particularly on CIC-IoT, which contains 1,048,575 samples spanning 34 classes.

To reduce computational complexity and improve training efficiency, we regrouped the 34 original CIC-IoT attack classes into 7 broader semantic categories and then downsampled each category by a factor of 10 (see Table 5). This reduced dataset size while keeping representative samples from every class. To ensure that this preprocessing did not distort the class balance, we compared the class distributions of the original and downsampled datasets using three standard statistical tests. The Chi-square test checks whether two distributions are statistically different in their frequency counts. The Jensen–Shannon Divergence measures how similar two probability distributions are, where values close to zero mean the distributions are effectively identical. Total Variation Distance measures the maximum difference in probability across all classes, where zero indicates no deviation at all. Our results showed Chi-square = 0.0098 (*p* = 1), JSD = 0.0043, and TVD = 0.0067, which together demonstrate that the downsampled dataset maintains the same overall class proportions as the original. In other words, regrouping and downsampling did not introduce any distributional shift. This also simplifies the classification task, since the model now distinguishes among 7 broader categories rather than 34 fine-grained subtypes, leading to an improvement in FL training accuracy from 87.78% on the original dataset to 99.05% on the consolidated version, although the other performance metrics remain the same (see Appendix A for details).

To evaluate real-time feasibility, we also measured end-to-end training time under different configurations. Wall-clock training times for all models are reported in Figure A2 in Appendix B. As shown, centralized training accumulates computation time much faster because all processing is handled by a single node. In contrast, both FL and FLDP distribute the workload across multiple fog clients, which leads to a slower increase in total training time. This trend becomes more pronounced as the number of fog clients increases, where models with 200–400 fog clients remain well below the centralized runtime. These measurements confirm that the proposed framework offers lower end-to-end training time and is practical for near-real-time IoT settings.

### 5.3. Effect of Non-IID Data on PP-HFFL Training

In the non-IID setting, we investigated the combined effects of class imbalance and class absence on PP-HFFL training. Both RT-IoT and CIC-IoT datasets are inherently imbalanced, and to preserve this characteristic in PP-HFFL, we applied a uniform label assignment (label split) to distribute all data across fog clients without performing any rebalancing. To simulate controlled class missingness per client, we limited the maximum number of classes assigned to each client as follows: for RT-IoT, {12,6,3,2}; for CIC-IoT, {7,3,2}. The single-class-per-client scenario was deliberately excluded, as it represents an extreme case that can introduce a positive-class issue [59]. Additionally, we varied the number of fog clients (10, 50, 100, 200, 400) to analyze the impact of fog client scale on PP-HFFL training accuracy.

Table 6 summarizes the performance of the global PP-HFFL-based IDS under varying degrees of non-IID class skew with client participation ration c_ration={0.5,0.25} for both the RT-IoT 2022 and CIC-IoT 2023 datasets. When clients maintain moderate class diversity (RT-IoT: 12 or 6 classes per client; CIC-IoT: 7 or 3 classes per client), PP-HFFL achieves high accuracy (typically above 95%) while also maintaining stable precision, recall, and F1 scores, indicating balanced detection behavior and minimal client drift. However, when class diversity is severely reduced (e.g., only 3 or 2 classes per client), the model becomes highly sensitive to the number of participating clients, since many classes are absent from local updates. For example, on CIC-IoT with 2 classes per client and for total 10 clients, the accuracy drops to 59.49% and the F1 score to 26.64, reflecting unstable decision boundaries and reduced recall. As the number of clients increases, global class coverage improves, enabling performance recovery; in the same setting with 100 clients, CIC-IoT accuracy rises to 95.86%, with the F1 score increasing to 47.09. A similar pattern is observed in RT-IoT, where accuracy falls to 13.10–27.57% for 2–3 classes per client at 10 clients, but improves to 82.73–93.60% at 400 clients. Moreover, reducing the client participation ratio lowers computation and communication cost, but multiple hierarchical rounds help preserve comparable performance trends. Overall, these results demonstrate that sufficient class diversity per client stabilizes hierarchical training, while increasing the number of clients mitigates extreme non-IID effects by restoring global class coverage, improving not only accuracy but also the stability of precision, recall, and F1 score.

### 5.4. Effect of Differential Privacy in PP-HFFL Accuracy

To ensure data privacy in our PP-HFFL framework, we integrate differential privacy into the client-side training using Opacus (version 1.5.4) [60], a PyTorch library designed for privacy-preserving deep learning. In this setup, each fog client applies the Gaussian noise mechanism with a noise_multiplier of 1.0, adding noise drawn from a normal distribution (standard deviation 1.0) to the clipped local gradients. This noise level, together with a max_grad_norm of 1.0, enforces bounded sensitivity by ensuring that no single IoT data point disproportionately influences the model update. We also set appropriate values of **ϵ** and **δ**, which provide a practical balance between privacy and model utility in IoT deployments. After local training in each communication round, fog clients upload these differentially private model updates to the PP-HFFL server for hierarchical aggregation.

Figure 2 illustrates the accuracy–privacy trade-off on test data in our PP-HFFL IDS when the client participation ratio is 0.5 and when DP is applied with **ϵ=5.0** and **$\delta =10^{-5}$** for moderate privacy protection. For the RT-IoT dataset (12 classes, 79 features), accuracy fluctuates during the early communication rounds—particularly with 200–400 fog clients—due to stronger DP noise, before stabilizing. The non-DP PP-HFFL model achieves roughly 95–99% accuracy as illustrated in Table 6, while the DP-enabled PP-HFFL converges to roughly 94–98% shown in Table 7, representing a drop of 1.0–5.0 percentage points. This sensitivity is largely attributable to the higher number of classes and input features, which amplifies the effect of DP noise on local updates. In comparison, the CIC-IoT dataset (7 classes, 46 features) with **ϵ=5.0** and **$\delta =10^{-5}$** exhibits smoother accuracy curves and a smaller reduction due to DP. The non-DP PP-HFFL model maintains above 98% accuracy as shown in Table 6, while DP reduces performance to approximately 96–98% shown in Table 7, a decrease of around 1.0–2.0 points.

*Epsilon (ϵ) Fine-Tuning in DP.* Table 7 shows that for the RT-IoT dataset, under identical DP settings and a client participation ratio of 0.25 and ϵ=5.0, the accuracy ranges from 95.02% to 98.49%. When we further tune ϵ∈{1.0,3.0,8.0,10.0} using 10 fog clients, the resulting accuracies exhibit only marginal variation-{98.08, 98.32, 98.46, 98.62, 98.66}. All other performance metrics follow similar patterns.

For the CIC-IoT dataset, using the same DP settings and a 0.25 participation ratio, accuracy ranges from 96.64% to 98.75%. Hyperparameter tuning with ϵ∈{1.0,3.0,8.0,10.0} (10 fog clients) again leads to negligible accuracy changes-{98.50,98.63,98.72,98.85,98.90}, with other metrics showing the same behavior.

Overall, the results show that while DP introduces a slight reduction in accuracy-more pronounced for datasets with higher non-IID characteristics or larger numbers of fog clients-both datasets still achieve high final accuracy. This demonstrates that differential privacy can be effectively integrated into PP-HFFL without significantly compromising performance, supporting its feasibility for real-world IoT intrusion detection. Furthermore, the simpler class structure and lower input dimensionality reduce susceptibility to DP noise, allowing the hierarchical PP-HFFL model to maintain strong performance across varying numbers of fog clients, participation ratios, and privacy parameter settings.

### 5.5. Personalized PP-HFFL

In the personalized PP-HFFL experiments, we evaluated two key scenarios: (i) improving a trained global model when it performs poorly on a fog node’s local data, under both non-DP, and DP settings; (ii) enabling a newly joined fog node to adapt the global PP-HFFL model using its local data.

For the first scenario, we selected a global PP-HFFL model with suboptimal performance. For the second scenario, we retrained the global model on the new client’s local data, analogous to transfer learning within the PP-HFFL context. Both scenarios were tested under non-DP and DP conditions. Figure 3a show the experimental results for the non-DP case, and the DP case in Figure 3b follows the same trend.

Figure 3a shows the effect of personalized PP-HFFL in the non-DP setting on the RT-IoT dataset using 100 fog clients, each with 2 classes and a 0.5 client participation ratio. The baseline global model accuracy was 31.87%. The green bars represent each client’s accuracy before personalization, while the orange bars show the accuracy after personalization, which improves to an average of 98.16% when regularization parameter $\lambda =1\times 10^{-3}$ (where higher λ value indicates more global alignment, and lower λ value indicates more personalization). Most clients experience substantial improvements-often reaching performance levels comparable to a centralized ML model trained on the full dataset. However, improvements are not perfectly uniform due to variations in client dataset sizes and label distributions, and some residual bias persists under extreme non-IID conditions. Overall, these results demonstrate that personalization significantly boosts local performance when the global PP-HFFL model performs suboptimally.

A similar trend is observed in the DP setting (Figure 3b), using the same parameters, where the baseline DP model achieves 33.12% accuracy and personalization increases the average performance to 98.16%. Personalization offers an even more noticeable improvement under DP, since DP-based PP-HFFL models typically degrade due to noise injection.

The CIC-IoT dataset does not require such personalization because, in all cases, the FL accuracy is already higher than 95%. For newly joined fog nodes, however, personalization allows the hierarchical global model to quickly adapt to their local data-functionally similar to transfer learning-while still maintaining strong privacy guarantees.

### 5.6. Statistical Reporting of Performance Metrics of PP-HFFL

To complement the qualitative analysis presented in the preceding sections and ensure the rigor, transparency, and reproducibility of our PP-HFFL experiments, we report detailed statistical summaries of model performance under both DP and non-DP conditions.

Mean performance metrics (e.g., global accuracy) are computed together with standard deviations across n=30 independent epochs (10, 20, …, 300). The 95% confidence interval (CI) for the mean accuracy is obtained as$${\mathrm{CI}}_{95\%}=\bar{x}\pm t_{n-1,\;0.975}\;\frac{s}{\sqrt{n}}$$
where $\bar{x}$ is the sample mean, *s* is the sample standard deviation, and $t_{29,\;0.975}\approx 2.045$. All figures include error bars (±1 SD or ±95% CI), and summary tables present the Mean ± SD together with the corresponding 95% CIs.

For statistical comparison between experimental configurations (e.g., $n_{\mathrm{client}}=10$ vs. $n_{\mathrm{client}}=50$), we perform paired *t*-tests or Wilcoxon signed-rank tests when normality assumptions are not met. The resulting *p*-values and significance levels are reported to indicate whether observed differences are statistically meaningful.

*DP-enabled Accuracy Analysis (RT-IoT Dataset).* Figure 2a presents the differential privacy (DP)-enabled PP-HFFL accuracy results for the RT-IoT dataset. As summarized in Table 8 and Table 9, accuracy declines as the number of fog clients increases-particularly beyond $n_{\mathrm{client}}=100$-reflecting the greater sensitivity of distributed training to DP noise. The pairwise Wilcoxon tests confirm statistically significant differences (p<0.01) among most client configurations, validating the accuracy–privacy trade-off discussed earlier.

*DP-enabled Accuracy Analysis (CIC-IoT Dataset).* Figure 2b depicts the corresponding DP-enabled PP-HFFL accuracy trends for the CIC-IoT dataset. As shown in Table 10 and Table 11, accuracy remains relatively stable across varying client counts, with only minor fluctuations. Both paired *t*-tests and Wilcoxon tests yield *p*-values below 0.05 for most comparisons, indicating significant, yet consistent performance differences. These findings confirm the robustness of DP-enabled learning even under moderate noise injection.

*Personalization Analysis.* To further validate the consistency of PP-HFFL performance across scenarios, Figure 3a,b and Table 12, Table 13, Table 14 and Table 15 summarize the statistical comparisons for non-DP and DP personalization experiments on the RT-IoT dataset. In both settings, accuracy improves dramatically after personalization-rising from roughly 32–33% before personalization to over 98% after. Paired *t*-tests and Wilcoxon tests yield extremely low *p*-values (e.g., $p<10^{-16}$), confirming that the observed improvements are statistically significant.

A paired *t*-test for the DP scenario yields t(99)=−21.31, $p=9.23\times 10^{-39}$, confirming the robustness and consistency of improvement even under privacy constraints. Error bars in Figure 2 and Figure 3 correspond to ±1 SD (or ±95% CI), reinforcing the transparency and reproducibility of the experimental findings.

Overall, these results demonstrate that (i) DP introduces statistically significant yet controlled reductions in model accuracy as shown in Table 8, Table 9, Table 10 and Table 11, and (ii) personalization provides statistically significant gains under both DP and non-DP conditions as demonstrated in Table 12, Table 13, Table 14 and Table 15—underscoring the stability, reproducibility, and effectiveness of the proposed PP-HFFL framework.

### 5.7. Discussion and Limitations

Our experiments on RT-IoT 2022 and CIC-IoT 2023 datasets highlight several key findings in the context of PP-HFFL-based IDS:The PP-HFFL IDS maintains near-centralized accuracy, precision, recall, and F1 score when fog clients retain sufficient class diversity. Performance drops sharply under extreme non-IID splits, though increasing the number of clients partially mitigates this by improving overall class coverage within the hierarchical federation.Integrating DP into PP-HFFL introduces a modest accuracy reduction (approximately 1.3–5.8 points), along with a higher computational cost. A more noticeable accuracy drop is observed for the higher-dimensional RT-IoT dataset. Nevertheless, both datasets achieve strong final accuracy as well as high precision, recall, and F1 scores, demonstrating that DP can be incorporated into PP-HFFL without severely compromising model utility.Personalization within PP-HFFL significantly enhances local model performance, particularly under DP. Many fog clients approach baseline centralized performance, and newly joined nodes can efficiently adapt the hierarchical global model via a transfer-learning-like mechanism, preserving privacy while improving local accuracy.

Limitations of the current PP-HFFL-based IDS study include the following:The current PP-HFFL framework applies client-side differential privacy to protect local data from an honest-but-curious server, but it does not address Byzantine attacks, where malicious fog nodes or IoT devices may send corrupted or adversarial model updates. Future work will integrate Byzantine-robust aggregation mechanisms with the DP components to simultaneously achieve both privacy and robustness guarantees.The current implementation assumes synchronous federated aggregation with fixed participation rates and static fog node assignments. In practice, IoT deployments experience dynamic node churn and intermittent connectivity. Future work will extend PP-HFFL to support asynchronous FL with partial participation, straggler handling, and staleness-aware aggregation. Investigating how heterogeneous computational and communication capacities affect convergence and privacy guarantees will also be important for real-world deployments.Although ϵ is tuned under a fixed δ, a more substantive limitation is the absence of any evaluation of privacy attacks—such as membership inference or model inversion-on the DP-enabled PP-HFFL framework, as well as the lack of integration and the assessment of secure aggregation algorithm.

## 6. Conclusions

This work presented a Privacy-Preserving Hierarchical Fog Federated Learning (PP-HFFL) framework for Intrusion Detection Systems (IDS), designed to overcome the limitations of resource-constrained IoT environments. By offloading model training and decision-making to fog nodes, the proposed approach alleviates computational burdens on IoT devices while simultaneously addressing scalability, privacy, and data heterogeneity challenges inherent in distributed IoT ecosystems. In PP-HFFL, a global model is trained at the cloud using the FedAvg algorithm, with fog nodes acting as federated clients. To safeguard sensitive information, client-side differential privacy mechanisms are incorporated into local training, protecting model updates from potential inference attacks. Additionally, model personalization is applied at the fog layer to fine-tune local models for each node, enabling adaptation to dynamic environments and seamless integration of newly joined nodes into the federated network.

The framework was experimentally validated using two benchmark IoT datasets-RT-IoT 2022 and CIC-IoT 2023. The evaluations demonstrate that PP-HFFL achieves performance comparable to centralized approaches, even under heterogeneous data distributions and varying client scales. Incorporating DP introduces a modest trade-off between privacy and performance, consistent with prior DP-FL studies [45,61]. Importantly, model personalization substantially enhances local performance, particularly in DP-enabled settings, ensuring that each fog node benefits from context-specific model refinement. These results confirm that PP-HFFL maintains privacy guarantees while providing high detection effectiveness across diverse fog computing environments.

Future research will focus on four key directions: (i) extending PP-HFFL to asynchronous FL with partial participation, dynamic node churn, and staleness-aware aggregation; (ii) integrating Byzantine-robust aggregation methods with DP and analyzing the accuracy–privacy–robustness trade-off under varying proportions of malicious clients; (iii) improving communication efficiency through techniques such as gradient compression, sparsification, and adaptive communication schedules to reduce bandwidth usage in large-scale IoT deployments; (iv) expanding the threat model to include stronger adversarial attacks such as model poisoning, inference attacks, and adversarial perturbations, evaluating PP-HFFL under these extended attack scenarios.

## Figures and Tables

**Figure 1 sensors-25-07296-f001:**
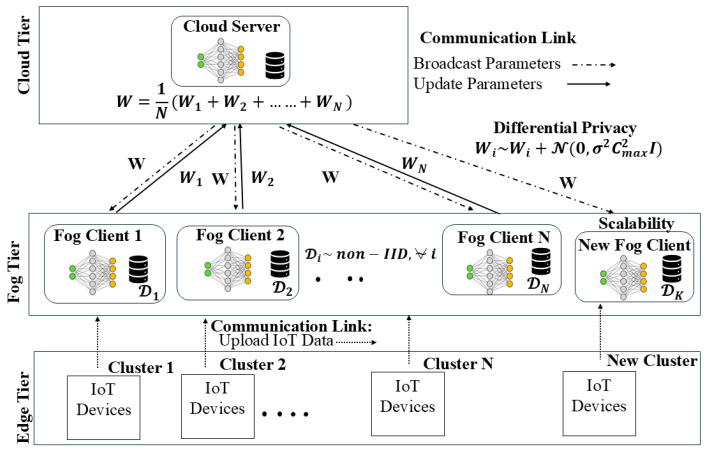
Privacy-Preserving Hierarchical Fog Federated Learning (PP-HFFL) architecture with cloud (global initialization/aggregation), fog (DP-enabled local training and optional personalization), and edge tiers (data collection and preprocessing) enabling privacy-preserving, scalable IoT IDS.

**Figure 2 sensors-25-07296-f002:**
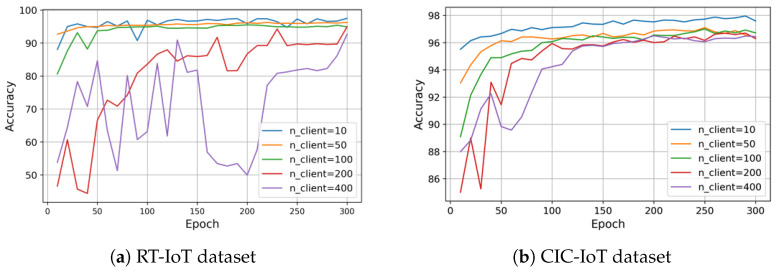
Accuracy of PP-HFFL global models on test data under DP with varying fog client counts. Across 300 communication rounds, DP reduces accuracy by approximately 1.71–5.78% for RT-IoT and 1.33–2.65% for CIC-IoT datasets, demonstrating the accuracy–privacy trade-off in privacy-preserving hierarchical federated learning.

**Figure 3 sensors-25-07296-f003:**
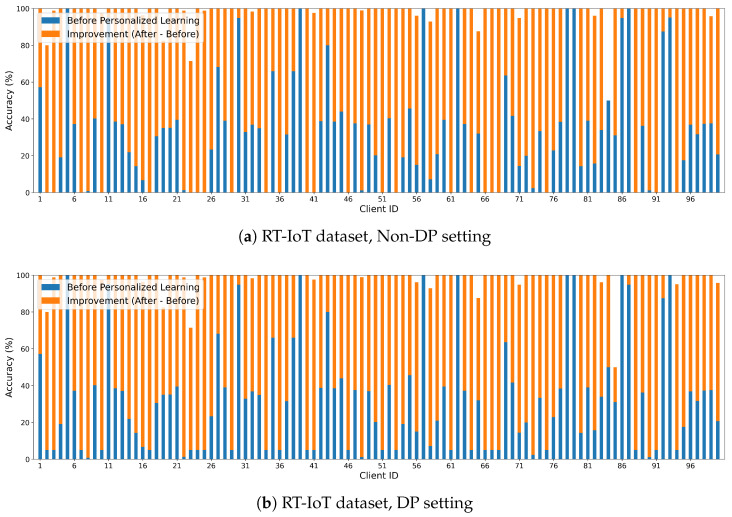
PP-HFFL: Local performance improvement of 100 fog clients after personalizing a pre-trained global FL model. (**a**) shows results in the non-DP setting, and (**b**) shows results under DP. Personalization significantly boosts client-level accuracy, particularly for nodes with few samples or highly skewed class distributions.

**Table 1 sensors-25-07296-t001:** Fog-based FL approaches for IDS. Summarizing contributions: non-IID handling, performance, scalability, and privacy considerations.

Ref	Year	Main Contribution	Non-IID Handling	Performance	Scalability	Privacy
[42]	2020	Fog-FL: hierarchical FL where fog nodes train/aggregate locally before cloud synchronization.	–	✔	–	–
[41]	2022	Proposed fog-client selection (random or resource-aware) in FL training, optimizing the performance.	–	✔	–	–
[39]	2023	Incorporates fog computing into FL to offload training from IoT devices, improving IDS performance.	–	✔	–	–
[43]	2023	Fog-based FL framework for IDS, leveraging fog-layer processing to enhance scalability and responsiveness.	–	✔	✔	–
[44]	2024	Proposed a privacy-preserving fog–federated IDS combining GAN-based data augmentation and differential privacy to address non-IID and adversarial data leakage.	✔	✔	–	✔
[47]	2025	Proposed an intelligent intrusion detection mechanism, FedACNN, by assisting CNN through the Federated Learning Mechanism.	–	✔	✔	✔
[40]	2025	Secured IoT via fog-layer FL, enabling collaborative IDS with privacy preservation.	–	✔	✔	✔

**Table 2 sensors-25-07296-t002:** Attack type and total number of data samples per class in RT-IoT 2022 and CIC-IoT 2023 datasets.

RT-IoT Dataset		CIC-IoT Dataset			
Attack Type	Count	Attack Type	Count	Attack Type	Count
DoSSYNHping	94,659	DDoSICMPFlood	161,281	DDoSUDPFlood	121,205
ThingSpeak	8108	DDoSTCPFlood	101,293	DDoSPSHACKFlood	92,395
ARPPoisoning	7750	DDoSSYNFlood	91,644	DDoSRSTFINFlood	90,823
MQTTPublish	4146	DDoSSynIPFlood	80,680	DoSUDPFlood	74,787
NmapUDPScan	2590	DoSTCPFlood	59,807	DoSSynFlood	45,207
NmapXMAStreesc	2010	BenignTraffic	24,476	MiraiGreethFlood	22,115
NmapOSDetection	2000	MiraiUdpplain	20,166	MiraiGreipFlood	16,952
NmapTCPScan	1002	DDoSICMPFrag	10,223	MITMArpSpoofing	7019
DDOSSlowloris	534	DDoSACKFrag	6431	DDoSUDPFrag	6431
Wiprobulb	253	DNSSpoofing	4034	ReconHostDisc	3007
MetasploitBF	37	ReconOSScan	2225	ReconPortScan	1863
NmapFINScan	28	DoSHTTPFlood	1680	VulnerabilityScan	809
		DDoSHTTPFlood	626	DDoSSlowLoris	493
		DictionaryBF	324	BrowserHijacking	140
		SqlInjection	122	CommandInjection	105
		BackdoorMalware	76	XSS	72
		ReconPingSweep	41	UploadingAttack	23

**Table 3 sensors-25-07296-t003:** Neural network architecture for PP-HFFL IDS.

Layer (Type)	Output	Param #
Input Layer (Linear)	(128)	d×128
BatchNorm1d	(128)	256
ReLU	(128)	0
Linear	(256)	128×256
BatchNorm1d	(256)	512
ReLU	(256)	0
Linear	(256)	256×256
BatchNorm1d	(256)	512
ReLU	(256)	0
Linear	(128)	256×128
BatchNorm1d	(128)	256
ReLU	(128)	0
Output Layer (Linear)	(*n*)	128×n

*d* = feature dimension; *n* = number of classes.

**Table 4 sensors-25-07296-t004:** Hyperparameters for PP-HFFL IDS training.

Name	Value
Aggregation algorithm	FedAvg
Total classes	12, 7
Input dimension	79, 46
Max training rounds	300
Local epochs per round	5
Batch size	64
Optimizer	SGD
Initial learning rate	0.03
Weight decay	1 $\times \;10^{-5}$
Noise multiplier	1.0
Max_grad_norm	1.0
Privacy parameter	5.0
Failure probability parameter	1 $\times \;10^{-5}$

Included DP parameters as well.

**Table 5 sensors-25-07296-t005:** Mapping the 34 CIC-IoT 2023 malware classes to 7 consolidated categories, with sample counts before and after downsampling, aligned with PP-HFFL.

New Class	Old Class	Count	Downsampled Count
Flood Attacks	DDoSICMPFlood, DDoSTCPFlood, DDoSUDPFlood, DoSTCPFlood, DoSUDPFlood, DoSSynFlood, DDoSPSHACKFlood, DDoSRSTFINFlood, DoSHTTPFlood, DDoSSYNFlood, DDoSSynIPFlood, DDoSHTTPFlood	921,428	92,143
Botnet/Mirai Attacks	MiraiGreethFlood, MiraiUDPplain, MiraiGreipFlood	59,233	5924
Benign	BenignTraffic	24,476	2448
Spoofing/MITM	DNSSpoofing, MITMArpSpoofing	11,053	1106
Reconnaissance	ReconHostDisc, ReconOSScan, ReconPortScan, ReconPingSweep	7136	714
Backdoors & Exploits	BackdoorMalware, UploadingAttack, BrowserHijacking, DictionaryBF	563	57
Injection Attacks	SqlInjection, CommandInjection, XSS	299	30

**Table 6 sensors-25-07296-t006:** Performance of the global PP-HFFL-based IDS on test data under non-IID settings, evaluated with different client participation ratios and varying numbers of clients, using the RT-IoT 2022 and CIC-IoT 2023 datasets.

			RT-IoT Dataset				CIC-IoT Dataset		
Client/Class	c_Ratio	Metric	12	6	3	2	7	3	2
10	0.5	Accuracy	99.17	95.93	27.57	13.10	98.77	98.27	59.49
		Precision	96.74	89.51	39.76	29.96	67.76	40.41	32.59
		Recall	94.30	87.40	32.42	35.98	63.97	44.91	33.81
		F1 Score	95.27	86.80	27.74	26.19	65.37	30.55	26.64
50	0.5	Accuracy	98.42	96.59	22.79	9.48	98.80	83.10	89.44
		Precision	78.94	88.15	65.46	32.62	62.20	73.94	60.56
		Recall	73.47	79.39	60.75	43.83	54.12	56.03	46.81
		F1 Score	75.52	82.77	55.25	27.21	55.59	58.70	47.56
100	0.5	Accuracy	98.40	98.47	69.09	31.87	98.75	98.69	95.86
		Precision	80.04	87.11	59.86	56.96	60.37	75.56	66.49
		Recall	72.91	81.63	61.06	47.01	56.80	53.07	45.05
		F1 Score	75.17	83.52	54.83	38.79	57.81	54.97	47.09
200	0.5	Accuracy	98.93	98.41	94.67	41.42	98.89	98.62	96.17
		Precision	79.95	79.46	65.62	58.75	59.12	60.39	44.74
		Recall	78.86	76.39	63.18	54.19	56.04	49.24	39.36
		F1 Score	79.25	77.23	62.38	49.39	56.90	49.87	34.48
400	0.5	Accuracy	95.67	98.30	93.60	82.73	98.45	98.61	96.53
		Precision	76.4	80.35	71.76	63.85	60.73	59.00	56.47
		Recall	71.23	73.22	56.14	50.94	48.57	46.22	42.44
		F1 Score	68.72	75.69	56.71	47.97	49.52	45.65	44.15
10	0.25	Accuracy	93.16	94.64	33.67	2.68	98.90	91.11	90.22
		Precision	93.16	87.50	58.72	2.72	63.54	27.08	45.97
		Recall	94.87	83.32	47.12	11.30	61.61	39.91	22.77
		F1 Score	93.80	82.36	43.47	2.43	62.29	24.29	22.25
50	0.25	Accuracy	99.12	96.56	23.37	10.22	98.97	98.04	98.33
		Precision	80.0	85.65	61.58	31.47	61.26	78.53	72.19
		Recall	80.0	7936	46.55	41.94	57.62	49.84	48.85
		F1 Score	79.86	81.41	39.90	22.39	58.69	52.56	49.28
100	0.25	Accuracy	98.33	98.93	17.52	33.50	98.75	98.66	96.82
		Precision	79.32	80.21	62.94	33.05	65.45	73.70	44.89
		Recall	72.17	78.20	57.12	44.54	56.31	54.20	44.16
		F1 Score	74.34	78.99	52.44	35.45	52.46	56.55	43.05
200	0.25	Accuracy	98.12	98.36	86.96	10.91	98.36	98.40	96.65
		Precision	78.12	80.32	61.32	34.67	54.33	57.66	49.59
		Recall	69.63	73.90	62.11	46.35	57.3	53.08	42.70
		F1 Score	70.08	76.26	59.04	34.07	43.71	54.41	39.47
400	0.25	Accuracy	98.97	98.40	93.90	50.09	96.64	98.66	97.43
		Precision	69.17	80.0	73.64	59.85	60.45	61.81	57.60
		Recall	68.26	73.05	63.49	62.15	56.63	47.30	42.30
		F1 Score	68.63	75.55	64.93	57.18	56.34	47.35	46.00

**Table 7 sensors-25-07296-t007:** Accuracy of the global PP-HFFL-based IDS on test data for different client participation ratios and different value of ϵ under DP, evaluated on the RT-IoT 2022 and CIC-IoT 2023 datasets.

		RT-IoT Dataset	CIC-IoT Dataset
Client	c_ratio, ϵ=5.0	12 Class	7 Class
10	0.5	98.46	98.72
50	0.5	98.42	98.79
100	0.5	98.40	98.75
200	0.5	98.06	98.36
400	0.5	94.13	96.01
10	0.25	98.49	98.70
50	0.25	98.42	98.75
100	0.25	98.33	98.75
200	0.25	98.12	98.36
400	0.25	95.02	96.64
10	1.0	98.08	98.50
10	3.0	98.32	98.63
10	5.0	98.46	98.72
10	8.0	98.62	98.85
10	10.0	98.66	96.90

**Table 8 sensors-25-07296-t008:** Descriptive statistics for DP-enabled PP-HFFL on RT-IoT dataset.

Configuration	Mean Accuracy	SD	95% CI
$n_{\mathrm{client}}=10$	0.9598	0.0201	[0.9523, 0.9673]
$n_{\mathrm{client}}=50$	0.9554	0.0078	[0.9525, 0.9583]
$n_{\mathrm{client}}=100$	0.9388	0.0309	[0.9273, 0.9504]
$n_{\mathrm{client}}=200$	0.8035	0.1423	[0.7503, 0.8566]
$n_{\mathrm{client}}=400$	0.7134	0.1356	[0.6628, 0.7641]

**Table 9 sensors-25-07296-t009:** Pairwise tests and significance for DP-enabled PP-HFFL on RT-IoT dataset.

Comparison	Test Type	*p*-Value	Significant?
10 vs. 50	Wilcoxon	0.0040	Yes
10 vs. 100	Wilcoxon	0.0000028	Yes
50 vs. 100	Wilcoxon	0.0000000019	Yes
200 vs. 400	Wilcoxon	0.0043	Yes

**Table 10 sensors-25-07296-t010:** Descriptive statistics for DP-enabled PP-HFFL on CIC-IoT dataset.

Configuration	Mean Accuracy	SD	95% CI
10	0.9725	0.0056	[0.9704, 0.9746]
50	0.9632	0.0082	[0.9602, 0.9663]
100	0.9579	0.0164	[0.9517, 0.9640]
200	0.9610	0.0300	[0.9487, 0.9733]
400	0.9640	0.0265	[0.9519, 0.9761]

**Table 11 sensors-25-07296-t011:** Pairwise tests and significance for DP-enabled PP-HFFL on CIC-IoT dataset.

Comparison	Test Type	*p*-Value	Significant?
10 vs. 50	Paired *t*-test	2.23 $\times \;10^{-13}$	Yes
10 vs. 100	Paired *t*-test	1.05 $\times \;10^{-7}$	Yes
50 vs. 100	Paired *t*-test	1.91 $\times \;10^{-3}$	Yes
200 vs. 400	Paired *t*-test	2.93 $\times \;10^{-1}$	No
10 vs. 50	Wilcoxon	1.73 $\times \;10^{-6}$	Yes
10 vs. 100	Wilcoxon	1.86 $\times \;10^{-9}$	Yes
50 vs. 100	Wilcoxon	3.90 $\times \;10^{-5}$	Yes
200 vs. 400	Wilcoxon	1.75 $\times \;10^{-2}$	Yes

**Table 12 sensors-25-07296-t012:** Descriptive statistics for non-DP before vs. after Personalization.

Configuration	Mean Accuracy	SD	95% CI
Before	0.3188	0.3127	[0.2567, 0.3808]
After	0.9816	0.0642	[0.9689, 0.9944]

**Table 13 sensors-25-07296-t013:** Statistical significance for non-DP before vs. after Personalization.

Comparison	Test Type	*p*-Value	Significant?
Before vs. After	Paired *t*-test	5.92 $\times \;10^{-38}$	Yes
Before vs. After	Wilcoxon	1.01 $\times \;10^{-16}$	Yes

**Table 14 sensors-25-07296-t014:** Descriptive statistics for DP before vs. after Personalization.

Configuration	Mean Accuracy	SD	95% CI
Before	0.3313	0.3004	[0.2717, 0.3909]
After	0.9816	0.0642	[0.9689, 0.9944]

**Table 15 sensors-25-07296-t015:** Statistical significance for DP before vs. after Personalization.

Comparison	Test Type	*p*-Value	Significant?
Before vs. After	Paired *t*-test	9.23 $\times \;10^{-39}$	Yes
Before vs. After	Wilcoxon	1.32 $\times \;10^{-16}$	Yes

## Data Availability

The datasets used in this study are publicly available and have been cited properly in the main paper.

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
