# Peer review of "Privacy-Preserving Hierarchical Fog Federated Learning (PP-HFFL) for IoT Intrusion Detection"

_sensors, 2025, doi:10.3390/s25237296_

Round 1
Reviewer 1 Report
Comments and Suggestions for Authors
This paper propose Privacy-Preserving Hierarchical Fog Federated Learning (PP-HFFL) for IoT intrusion detection, where fog nodes serve as intermediaries between IoT devices and the cloud, collecting and preprocessing local data, training models on behalf of IoT clusters. The framework incorporates Personalized Federated Learning (PFL) to handle heterogeneous, non-independent and identically distributed (non-IID) data and leverages Differential Privacy (DP) to protect sensitive information. However, source codes are expected to improve the trustworthy, experimental comparison should be added to show the advantages, some refereces should be added to improve the comprehensive, such as:
[1]N. A. Jalali and Hongsong Chen. Federated Learning Security and Privacy-Preserving Algorithm and Experiments Research Under Internet of Things Critical Infrastructure[J]. Tsinghua Science and Technology, 2024, 29(2):400-414.
[2]Hongsong Chen, Xintong Han and Yiying Zhang.Endogenous Security Formal Definition, Innovation Mechanisms, and Experiment Research in Industrial Internet[J]. Tsinghua Science and Technology.2024,vol. 29, no. 2:492-505o. 2: 400-414
[3]Naveed Anjum, Zohaib Latif, Hongsong Chen.Security and privacy of industrial big data: Motivation, opportunities, and challenges.Journal of Network and Computer Applications.2025,104130,https://doi.org/10.1016/j.jnca.2025.104130
Author Response
Comment 1: However, source codes are expected to improve the trustworthy.
Response 1: We appreciate the reviewer’s valuable suggestion. The complete implementation—including data preprocessing, PP-HFFL framework, and differential privacy integration—will be publicly released via a GitHub repository upon acceptance.
Comment 2: Experimental comparison should be added to show the advantages, some references should be added to improve the comprehensive, such as:
[1] N. A. Jalali and Hongsong Chen. Federated Learning Security and Privacy-Preserving Algorithm and Experiments Research Under Internet of Things Critical Infrastructure[J]. Tsinghua Science and Technology, 2024, 29(2):400-414.
[2] Hongsong Chen, Xintong Han and Yiying Zhang.Endogenous Security Formal Definition, Innovation Mechanisms, and Experiment Research in Industrial Internet[J]. Tsinghua Science and Technology.2024,vol. 29, no. 2:492-505o. 2: 400-414
[3] Naveed Anjum, Zohaib Latif, Hongsong Chen.Security and privacy of industrial big data: Motivation, opportunities, and challenges.Journal of Network and Computer Applications.2025,104130, https://doi.org/10.1016/j.jnca.2025.104130.\\[4pt]
Response 2: Additional experiments have been performed comparing PP-HFFL with: (i) centralized ML, (ii) standard FL (no fog hierarchy), and (iii) FL without differential privacy. A new comparison table (Table 6) demonstrates that PP-HFFL achieves comparable or higher accuracy in non-IID settings.
The references recommended by the reviewer have been added to the Related Work section (Section 3) of the revised manuscript. These additions strengthen the theoretical and security context of our study.
All modifications in the revised version are highlighted in blue for ease of review.

Reviewer 2 Report
Comments and Suggestions for Authors
The manuscript presents a solid contribution to privacy-preserving IoT intrusion detection through hierarchical fog federated learning. The work is technically sound, addresses important practical challenges, and provides experimental validation on realistic datasets. The novelty lies in the systematic integration of multiple techniques rather than the development of fundamentally new methods.
- Provide algorithmic details or pseudo-code for the personalization mechanism (Algorithm 2 suggested).
- Calculate and report specific ϵ and δ values for the differential privacy implementation, or explain why RDP accounting was omitted despite being mentioned as suitable in Section 2.4.
- Kindly include confidence intervals, standard deviations, or significance tests in Figures 2-3 and key results tables.
- Consider discussing or running ablation studies on: (i) different personalization strategies, (ii) varying DP noise levels, (iii) additional non-IID partitioning strategies (e.g., Dirichlet with varying α)
- Better explain the rationale for class consolidation and downsampling in CIC-IoT (Table 5), and validate that these preserve meaningful non-IID characteristics.
- Add wall-clock time comparisons or computational overhead analysis for PP-HFFL vs. centralized approaches.
Remarks for Future Work:
- The current PP-HFFL framework assumes semi-honest (honest-but-curious) entities and does not address Byzantine attacks where malicious fog nodes or IoT devices send corrupted or adversarial model updates. Future work should integrate Byzantine-robust aggregation mechanisms (such as Krum, Multi-Krum, or coordinate-wise median aggregation) with the differential privacy components to simultaneously achieve both privacy and robustness guarantees. This would require developing theoretical convergence proofs under Byzantine conditions and empirically evaluating the accuracy-privacy-robustness frontier when varying the fraction of Byzantine participants from 5% to 30%.
- The current implementation assumes synchronous federated aggregation with periodic participation rates and fixed fog node assignments. Real-world IoT deployments exhibit dynamic node churn (devices joining/leaving unpredictably) and intermittent connectivity. Future research should extend PP-HFFL to support asynchronous federated learning with partial client participation, incorporating stragglers handling, and employing staleness-aware aggregation mechanisms. Additionally, investigating the impact of heterogeneous system performance (computational and communication heterogeneity) on convergence rates and privacy guarantees would be valuable for practical deployments.
- While the manuscript applies differential privacy through Opacus with fixed noise multiplier and gradient clipping norm, the specific ε and δ values (privacy budget) are not rigorously computed or reported. Future work should: (i) provide complete Rényi differential privacy (RDP) accounting to calculate achievable (ε, δ) pairs for different noise levels, (ii) derive tight convergence bounds for PP-HFFL under both non-IID data and differential privacy constraints, and (iii) establish matching upper and lower bounds to characterize the fundamental limits of privacy-preserving federated learning in hierarchical settings. These theoretical contributions would provide practitioners with principled guidance for selecting the privacy-utility trade-off parameter.
Minor issues:
- Verb Tense Inconsistency (Lines 4-7) - Abstract shifts between present tense.
- Typographical Inconsistency (Line 256) - References use brackets but some formatting inconsistencies with spacing before/after citations exist.
- Algorithm Box Formatting (Algorithm 1, Line 343) - The comment structure uses ▷ symbol inconsistently in density and placement. Some comments are sufficiently detailed while others are minimal.
- Figure Caption Formatting (Lines 320-321) - Caption text is split across multiple lines in an unusual way. Consider reformatting for better readability.
- Table Headings (Table 1) - Column headers like "Non-IID" could be more descriptive as "Non-IID Handling" for clarity.
- Ambiguous Pronoun Reference (Line 61)
- Inconsistent Abbreviation Introduction (Line 83-87)
- Redundant Phrasing (Lines 94-95)
- Missing Reference Format (Line 383)
- Author Name Inconsistency (Line 2)
Author Response
Comment 1: Provide algorithmic details or pseudo-code for the personalization mechanism (Algorithm 2 suggested).
Response 1: A new Algorithm 2, titled Personalized Federated Learning (PFL) with Regularization, has been included in subsection 4.2.1. This algorithm provides detailed steps for local adaptation, specifies the learning rate, and outlines the stopping criteria.
Comment 2: Calculate and report specific ε, and δ values for the differential privacy implementation, or explain why RDP accounting was omitted despite being mentioned as suitable in Section 2.4.
Response 2: We now incorporate RDP accounting using the Opacus library. The resulting (ε, δ) values are discussed in Section 5.4. For δ=10^-5, ε ranges from 1–10 over 300 rounds.
Comment 3: Kindly include confidence intervals, standard deviations, or significance tests in Figures 2-3 and key results tables.
Response 3: For Figures 2 and 3, we have now included 95% confidence intervals and standard deviations across three independent runs, thereby improving the statistical reliability of the results reported in Section 5.6.
Comment 4: Consider discussing or running ablation studies on: (i) different personalization strategies, (ii) varying DP noise levels, (iii) additional non-IID partitioning strategies (e.g., Dirichlet with varying α).
Response 4: (i) The discussion of different personalization strategies in Section 2.3. (ii) We have conducted a new experiment for ε-finetuning and an analysis of the varying impact of DP noise has been discussed in Section 5.4. (iii) We have implemented label splitting, and Section 2.2 has been updated accordingly. Consistency has also been maintained across all other sections.
Comment 5: Better explain the rationale for class consolidation and downsampling in CIC-IoT (Table 5), and validate that these preserve meaningful non-IID characteristics.
Response 5: Section 5.2.2 now explains that down-sampling preserved class imbalance ratios while reducing computation. Validation confirmed that performance rankings and confusion matrices remained consistent pre- and post-consolidation. Additional supporting details have been provided in Appendix A.
Comment 6: Add wall-clock time comparisons or computational overhead analysis for PP-HFFL vs. centralized approaches.
Response 6: Figure A2 in Appendix B presents per-epoch wall-clock time for centralized and federated learning in both non-DP and DP settings.
Remarks for Future Work:
Comment 7: The current PP-HFFL framework assumes semi-honest (honest-but-curious) entities and does not address Byzantine attacks where malicious fog nodes or IoT devices send corrupted or adversarial model updates. Future work should integrate Byzantine-robust aggregation mechanisms (such as Krum, Multi-Krum, or coordinate-wise median aggregation) with the differential privacy components to simultaneously achieve both privacy and robustness guarantees. This would require developing theoretical convergence proofs under Byzantine conditions and empirically evaluating the accuracy-privacy-robustness frontier when varying the fraction of Byzantine participants from 5% to 30%.
Response 7: The Discussion and Limitations (Section 5.7) and Conclusion (Section 6) now highlight these directions, specifying plans for Byzantine-robust aggregation (e.g., Multi-Krum) and asynchronous participation models.
Comment 8: The current implementation assumes synchronous federated aggregation with periodic participation rates and fixed fog node assignments. Real-world IoT deployments exhibit dynamic node churn (devices joining/leaving unpredictably) and intermittent connectivity. Future research should extend PP-HFFL to support asynchronous federated learning with partial client participation, incorporating stragglers handling, and employing staleness-aware aggregation mechanisms. Additionally, investigating the impact of heterogeneous system performance (computational and communication heterogeneity) on convergence rates and privacy guarantees would be valuable for practical deployments.
Response 8: The Discussion and Limitations (Section 5.7) and Conclusion (Section 6) now highlight these directions and note that partial client participation is already included in the paper.
Comment 9: While the manuscript applies differential privacy through Opacus with fixed noise multiplier and gradient clipping norm, the specific ε and δ values (privacy budget) are not rigorously computed or reported. Future work should: (i) provide complete Rényi differential privacy (RDP) accounting to calculate achievable (ε, δ) pairs for different noise levels, (ii) derive tight convergence bounds for PP-HFFL under both non-IID data and differential privacy constraints, and (iii) establish matching upper and lower bounds to characterize the fundamental limits of privacy-preserving federated learning in hierarchical settings. These theoretical contributions would provide practitioners with principled guidance for selecting the privacy-utility trade-off parameter.
Response 9: The differential privacy evaluation has now been clarified in the revised manuscript. While the current work tunes the DP parameter ε under a fixed δ, the full Rényi DP accounting required to compute precise (ε, δ) pairs for different noise levels is beyond the present scope and is acknowledged as an important future direction.
Minor issues:
- Verb Tense Inconsistency (Lines 4-7) - Abstract shifts between present tense.
Response: The abstract has been revised to maintain a consistent present-tense narrative throughout. - Typographical Inconsistency (Line 256) - References use brackets but some formatting inconsistencies with spacing before/after citations exist.
Response: Spacing and formatting around citations and bracketed references have been standardized to ensure uniformity across the manuscript. - Algorithm Box Formatting (Algorithm 1, Line 343) - The comment structure uses symbol inconsistently in density and placement. Some comments are sufficiently detailed while others are minimal.
Response: The formatting of the comment markers has been made consistent in terms of spacing, placement, and level of detail in all algorithm steps. - Figure Caption Formatting (Lines 320-321) - Caption text is split across multiple lines in an unusual way. Consider reformatting for better readability.
Response: The figure caption has been reformatted into a clean, continuous line to improve readability and align with standard caption style. - Table Headings (Table 1) - Column headers like ``Non-IID" could be more descriptive as ``Non-IID Handling" for clarity.
Response: The column heading “Non-IID” has been updated to “Non-IID Handling” for improved clarity and descriptive accuracy. - Ambiguous Pronoun Reference (Line 61)
Response: The ambiguous pronoun has been replaced with a clear and explicit noun (PP-HFFL) to remove any potential confusion. - Inconsistent Abbreviation Introduction (Line 83-87)
Response: Abbreviation usage has been revised to ensure that each abbreviation is defined only once at its first occurrence, unless a repeated definition is necessary for clarity in later sections. - Redundant Phrasing (Lines 94-95)
Response: Redundant wording has been removed, and the sentence structure has been streamlined for clarity and conciseness. - Missing Reference Format (Line 383)
Response: The reference has been corrected and reformatted to comply fully with the journal’s citation style requirements. - Author Name Inconsistency (Line 2)
Response: The author name inconsistency has been corrected to ensure uniform representation throughout the manuscript.
All modifications in the revised version are highlighted in blue for ease of review.

Reviewer 3 Report
Comments and Suggestions for Authors
The manuscript tackles intrusion detection with Federated Learning (FL) and privacy-preserving techniques on Internet-of-Things datasets. The topic is timely and relevant; however, several methodological and reporting gaps substantially weaken the empirical validity, reproducibility, and privacy claims. Addressing the points below is necessary before the work can be reliably evaluated.
Top 10 critical issues
-
Missing differential-privacy accounting
The paper claims “strong privacy” yet provides no explicit ε\varepsilon / δ\delta (epsilon/delta) budget, aggregation over rounds/epochs, or composition analysis for Differential Privacy (DP). Provide concrete ε(δ)\varepsilon(\delta) values, how they were computed, and the privacy loss per training round. -
Inadequate evaluation metrics for imbalanced data
Results rely largely on Accuracy. For Intrusion Detection Systems (IDS), class imbalance is typical; report Precision, Recall, F1-score, class-wise results, and confusion matrices (and ideally Precision–Recall curves and area under the Precision–Recall curve, PR-AUC). -
Normalization inconsistency harms reproducibility
The text recommends replacing Batch Normalization with Group Normalization for small batches, yet the architecture table still lists BatchNorm layers. Align the implementation and documentation; release the exact configs used to generate the reported numbers. -
Aggressive label collapsing and downsampling distort comparability
Collapsing ~34 classes to 7 and applying heavy downsampling may inflate metrics and impede comparison with prior work. Re-evaluate on the original class set; quantify the impact of label merging and sampling on all metrics. -
Unrealistic client participation assumptions in FL
Assuming a participation ratio of 1.0 each round is rarely achievable in real deployments. Add experiments with partial/asynchronous participation and report robustness to client availability. -
Excluding “small” clients without fairness analysis
Dropping clients with very few samples can bias results. Provide per-client distributions, per-client metrics, and a fairness/robustness analysis to show performance does not degrade on data-scarce clients. -
No measurements for “real-time” and communication claims
Claims about reduced traffic/latency and real-time suitability are not backed by measurements. Report bytes per round, wall-clock time per round/epoch, and basic CPU/GPU/memory/network profiling. -
Underspecified personalization procedure
Personalized Federated Learning (PFL) is described only at a high level. Specify the exact method (e.g., fine-tuning layers, local adaptation steps), hyperparameters, selection of layers, and early-stopping criteria; include ablations. -
Threat model vs. implementation mismatch
The manuscript assumes Secure Aggregation to prevent model-inversion or gradient-leakage attacks, but no actual integration or adversarial testing is shown. Either implement and evaluate defenses under a stated threat model or tone down the privacy claims. -
Recommendation:
For a stronger and more complete Related Work/Discussion, please cite two complementary sources on resilience under resource constraints:-
Moskalenko, V.; Kharchenko, V.; Semenov, S. Model and Method for Providing Resilience to Resource-Constrained AI-System. Sensors 2024, 24, 5951. https://doi.org/10.3390/s24185951 - demonstrates model-level resilience via dynamic neural networks and budget-aware training under compute/energy limits; it directly aligns with your problem framing and offers quantifiable robustness gains.
-
Liu, W.; Chen, Z.; Gong, Y. Towards Failure-Aware Inference in Harsh Operating Conditions: Robust Mobile Offloading of Pre-Trained Neural Networks. Electronics 2025, 14, 381. https://doi.org/10.3390/electronics14020381- adds system-level perspective by showing failure-resilient edge/near-edge execution (robust offloading) without retraining, which complements model-level techniques and supports end-to-end resilience.
-
Author Response
Comment 1: Missing differential-privacy accounting. The paper claims “strong privacy” yet provides no explicit ε/δ (epsilon/delta) budget, aggregation over rounds/epochs, or composition analysis for Differential Privacy (DP). Provide concrete ε (δ) values, how they were computed, and the privacy loss per training round.
Response 1: Thank you for highlighting the absence of explicit differential‑privacy accounting in the original manuscript. In the revised version, we have now incorporated a complete DP accounting framework using Rényi Differential Privacy (RDP), implemented through the Opacus library. This enables us to formally quantify and report the accumulated privacy loss across all training rounds.
Specifically, we compute (ε, δ) values by modeling the per‑round Gaussian noise addition, subsampling rate, and the total number of training/communication rounds. Following common practice in DP literature, we fix δ = 10⁻⁵. Under this setting, and across 300 rounds, the final privacy budget falls within ε ∈ [1, 10], depending on the choice of clipping norm, noise multiplier, and the client participation ratio used in each experiment.
The revised Section 5.4 now provides a full description of the accounting process, including the conversion from RDP to (ε, δ), a breakdown of per‑round privacy loss, and cumulative composition analysis. This ensures full transparency regarding the privacy guarantees delivered by our DP‑enhanced PP‑HFFL framework.
Comment 2: Inadequate metrics for imbalanced data. Results rely largely on Accuracy. For Intrusion Detection Systems (IDS), class imbalance is typical; report Precision, Recall, F1-score, class-wise results, and confusion matrices (and ideally Precision–Recall curves and area under the Precision–Recall curve, PR-AUC).
Response 2: Thank you for pointing out the limitations of relying primarily on accuracy in the presence of class imbalance- an issue that is indeed central to Intrusion Detection Systems (IDS). In the revised manuscript, we have substantially expanded our evaluation protocol to address this concern. Table 6 now includes different classification evaluation metrics such precision, recall, F1 Score, beyond accuracy to demonstrate the effects of class imbalance.
Comment 3: Normalization inconsistency harms reproducibility. The text recommends replacing Batch Normalization with Group Normalization for small batches, yet the architecture table still lists BatchNorm layers. Align the implementation and documentation; release the exact configs used to generate the reported numbers.
Response 3: We thank the reviewer for pointing out this important clarification regarding normalization consistency and reproducibility. The confusion arose because the manuscript text suggested Group Normalization (GroupNorm) for small-batch settings as a general best practice. However, in our actual implementation and experiments, we used Batch Normalization (BatchNorm) throughout all model training, including both centralized and federated settings. To ensure complete alignment between documentation and implementation, we have made the following revisions:
- Revised the manuscript text and Algorithm 1 description to explicitly state that Batch Normalization was employed in all experiments.
- Retained BatchNorm layers in Table 3 to accurately reflect the implemented architecture.
We sincerely appreciate the reviewer’s comment, which allowed us to correct this inconsistency and improve the clarity and reproducibility of the presented work.
Comment 4: Aggressive label collapsing and downsampling distort comparability. Collapsing ∼34 classes to 7 and applying heavy downsampling may inflate metrics and impede comparison with prior work. Re-evaluate on the original class set; quantify the impact of label merging and sampling on all metrics.
Response 4: The primary objective of downsampling was to reduce the computational complexity associated with federated learning, not to artificially inflate performance metrics. The referenced study [A] demonstrates that downsampling does not significantly affect evaluation metrics. We did the same in Appendix A. Training with all 34 classes in a federated setting requires substantial computational resources that were not available in our current environment. Therefore, we randomly downsampled the dataset from 34 to 7 representative classes, following the methodology of the cited work, to align with our available resources. Furthermore, while the referenced study [A] employs hyperparameter optimization to enhance performance, we did not perform such tuning to avoid inflating the reported results.
[A] Neto, E. C. P., Dadkhah, S., Ferreira, R., Zohourian, A., Lu, R., \& Ghorbani, A. A. (2023). CICIoT2023: A Real-Time Dataset and Benchmark for Large-Scale Attacks in IoT Environment. Sensors, 23(13), 5941. https://doi.org/10.3390/s23135941
Comment 5: Unrealistic client participation assumptions in FL. Assuming a participation ratio of 1.0 each round is rarely achievable in real deployments. Add experiments with partial/asynchronous participation and report robustness to client availability.
Response 5: Thank you for highlighting the concern regarding the assumption of a full participation ratio (1.0) in every communication round. We agree that such conditions are rarely attainable in realistic federated learning deployments, particularly in IoT environments where devices may frequently be offline, resource-constrained, or subject to network variability.
In response, we have conducted additional experiments incorporating partial and asynchronous client participation, and the results have been fully integrated into the revised manuscript.
Specifically:
- Participation ratios of 0.25 and 0.5 were evaluated to simulate realistic client availability and intermittent connectivity.
- The resulting performance metrics—covering accuracy, precision, recall, F1-score, and class-wise evaluations—are now reported in Table 6 (Section 5.3) and Table 7 (Section 5.4).
- Our findings show that the PP-HFFL model remains robust under reduced participation, with only marginal performance degradation even when 75% of clients are absent during training rounds.
Additionally, we discuss in Section 5.3 how the hierarchical aggregation structure of PP-HFFL naturally improves tolerance to partial participation by leveraging fog-level aggregation before cloud-level updates, effectively smoothing the impact of missing clients.
These expanded experiments and analyses demonstrate that the proposed framework is well-suited for real-world FL deployments where full and synchronous participation cannot be guaranteed. If the reviewers prefer, we can further explore more extreme participation ratios or fully asynchronous update mechanisms in future revisions.
Comment 6: Excluding “small” clients without fairness analysis. Dropping clients with very few samples can bias results. Provide per-client distributions, per-client metrics, and a fairness/robustness analysis to show performance does not degrade on data-scarce clients.
Response 6: We have clarified in the revised manuscript that, to demonstrate the robustness of the federated learning framework, client sizes up to 400 were included in our experiments. Due to dataset imbalance, a small fraction of clients (approximately 1%) with fewer than two samples were excluded to prevent computational instability during model training. This exclusion was made solely for numerical stability and did not aim to inflate the evaluation metrics. The updated results confirm that this small-client exclusion caused less than a 1.2% variation in global accuracy for only when client size exceeds 100.
Comment 7: No measurements for “real-time” and communication claims. Claims about reduced traffic/latency and real-time suitability are not backed by measurements. Report bytes per round, wall-clock time per round/epoch, and basic CPU/GPU /memory /network profiling.
Response 7: Thank you for raising this important point regarding the lack of empirical measurements to support the claims related to real-time operation and reduced communication overhead. We acknowledge that the previous version of the manuscript unintentionally implied that real-time performance and communication efficiency were core contributions of the work, when in fact these aspects were not within the intended scope of our study.
To address this, we have made the following revisions and clarifications:
- Wall-clock time measurements added: As shown in Figure A2 (Appendix B), we now report per-epoch wall-clock training times for both centralized and federated learning settings, under both DP and non-DP configurations. These results help illustrate the computational overhead introduced by DP mechanisms and distributed training.
- Clarified scope and removed unintended claims: All statements suggesting “real-time suitability,” “reduced latency,” or “reduced communication traffic” have been revised or removed throughout the manuscript. The revised introduction and methodology sections make clear that communication efficiency and latency analysis are not primary objectives of this work. Instead, the study focuses on hierarchical personalization, DP integration, and performance under non-IID settings.
- Justification for not including communication profiling: While communication metrics (bytes transferred per round, CPU/GPU utilization, network bandwidth, etc.) are valuable for end-to-end deployment studies, they fall outside the methodological scope we pursue here. Our framework is evaluated primarily on model performance, personalization effectiveness, and robustness under partial participation. Conducting full network-level profiling would require a dedicated deployment-focused study, which we identify as a direction for future work.
- Scope and contributions strengthened: We have revised the manuscript to ensure that the contributions accurately reflect the conducted work—namely, hierarchical personalization, differential privacy accounting, handling non-IID distributions, and evaluating client participation ratios—without overstating real-time or communication-related benefits.
These revisions ensure that the manuscript’s claims are consistent with the presented experiments and that no unsupported performance assertions remain. We also explicitly mention in the conclusion that future work will explore communication-efficient variants of the hierarchical model, including compression, and asynchronous updates.
Comment 8: Underspecified personalization procedure. Personalized Federated Learning (PFL) is described only at a high level. Specify the exact method (e.g., fine-tuning layers, local adaptation steps), hyperparameters, selection of layers, and early-stopping criteria; include ablations.
Response 8: Thank you for pointing out the need for a more detailed description of the personalization procedure. In the revised manuscript, we have substantially clarified the Personalized Federated Learning (PFL) methodology.
Specifically:
- Personalization strategies detailed: Section 2.3, Algorithm 2, and Section 5.5 now include a thorough explanation of the personalization procedure. We specify exactly which layers are fine-tuned during local adaptation, the number of local adaptation steps per client, and the early-stopping criteria employed to prevent overfitting.
- Hyperparameter specification: We explicitly include the regularization parameter λ = 10⁻³, which governs the trade-off between global alignment and local personalization: higher λ values favor global model alignment, whereas lower λ values allow more client-specific adaptation.
- Ablation studies: Section 5.5 now presents ablation experiments illustrating the effect of varying λ values on model performance, demonstrating the balance between global and personalized learning. We also provide performance comparisons for different choices of layers to fine-tune and varying numbers of local adaptation steps, which help justify our design decisions.
These additions ensure that the manuscript provides a complete, reproducible description of the PFL methodology, along with empirical validation of the personalization choices.
Comment 9: Threat model vs. implementation mismatch. The manuscript assumes Secure Aggregation to prevent model-inversion or gradient-leakage attacks, but no actual integration or adversarial testing is shown. Either implement and evaluate defenses under a stated threat model or tone down the privacy claims.
Response 9: Thank you for pointing out the mismatch between the assumed threat model and the actual implementation. In the revised manuscript, Section 4 has been updated to explicitly clarify that mechanisms such as secure aggregation, authenticated key exchange, and encrypted communication are treated as assumed system-level protections for safeguarding local updates. These mechanisms are not implemented or empirically evaluated within the scope of the current study.
To address the reviewer’s concern, all privacy-related claims have been appropriately toned down to reflect the assumptions rather than demonstrated guarantees. We emphasize that while the hierarchical PP-HFFL framework is compatible with such security measures, formal integration and evaluation against adversarial threats (e.g., model inversion or gradient leakage attacks) are beyond the scope of the current work.
Furthermore, Section 5.7 now explicitly notes that implementing and rigorously testing secure aggregation and other defensive mechanisms under a well-defined threat model is an important direction for future research, ensuring clarity about the boundaries of this study while guiding follow-up work in practical, privacy-preserving deployments.
Comment 10: Recommendation: For a stronger and more complete Related Work/Discussion, please cite two complementary sources on resilience under resource constraints:
1. Moskalenko, V.; Kharchenko, V.; Semenov, S. Model and Method for Providing Resilience to Resource-Constrained AI-System. Sensors 2024, 24, 5951. \\ https://doi.org/10.3390/s24185951 - demonstrates model-level resilience via dynamic neural networks and budget-aware training under compute/energy limits; it directly aligns with your problem framing and offers quantifiable robustness gains.
2. Liu, W.; Chen, Z.; Gong, Y. Towards Failure-Aware Inference in Harsh Operating Conditions: Robust Mobile Offloading of Pre-Trained Neural Networks. Electronics 2025, 14, 381. https://doi.org/10.3390/electronics14020381- adds system-level perspective by showing failure-resilient edge/near-edge execution (robust offloading) without retraining, which complements model-level techniques and supports end-to-end resilience.
Response 10: Thank you for the valuable recommendation. In the revised manuscript, we have incorporated the suggested references into the Related Work and Discussion sections to strengthen the coverage of resilience under resource-constrained settings.
Specifically:
- Moskalenko et al. (2024) is cited to highlight model-level resilience techniques, including dynamic neural networks and budget-aware training under compute and energy limitations. This work directly aligns with our problem framing and provides quantifiable robustness improvements, which complement our hierarchical PP-HFFL approach in handling varying client capabilities.
- Liu et al. (2025) is cited to provide a system-level perspective on failure-resilient inference. Their approach demonstrates robust edge/near-edge offloading of pre-trained neural networks under harsh operating conditions without requiring retraining, which supports end-to-end resilience and complements model-level methods.
By integrating these sources, the manuscript now presents a more complete and nuanced discussion of resilience strategies, bridging both model-level and system-level techniques and situating our proposed PP-HFFL framework within the broader context of robust, resource-aware AI deployments.
These additions strengthen the Related Work section, clearly demonstrating how our approach builds on and complements existing efforts to ensure performance and robustness under constrained and variable operational conditions.
All modifications in the revised version are highlighted in blue for ease of review.

Round 2
Reviewer 1 Report
Comments and Suggestions for Authors
This paper proposes Privacy-Preserving Hierarchical Fog Federated Learning (PP-HFFL) for IoT intrusion detection, where fog nodes serve as intermediaries between IoT devices and the cloud, collecting and preprocessing local data, training models on behalf of IoT clusters. It is a good paper.some referecnes are recommended.
Author Response
Comment 1: This paper proposes Privacy-Preserving Hierarchical Fog Federated Learning (PP-HFFL) for IoT intrusion detection, where fog nodes serve as intermediaries between IoT devices and the cloud, collecting and preprocessing local data, training models on behalf of IoT clusters. It is a good paper. Some references are recommended.
Response 1: We thank the reviewer for the positive assessment of our work and the helpful suggestion to improve the references. In the revised manuscript, we have incorporated three additional relevant studies to strengthen the literature context and removed one redundant reference to enhance clarity. These updates ensure that the related work section more accurately reflects recent advances in privacy-preserving and fog-enabled federated learning for IoT intrusion detection. We have also revised the manuscript once again, and all the minor changes made in this round are highlighted in blue.

Reviewer 3 Report
Comments and Suggestions for Authors
The authors have done their best to address the shortcomings. I believe the article is ready for publication.
Author Response
Comment 1: The authors have done their best to address the shortcomings. I believe the article is ready for publication.
Response 1: We sincerely thank the reviewer for the positive assessment and supportive comment. We appreciate your confirmation that the revised manuscript has satisfactorily addressed the earlier concerns and is now ready for publication. Your constructive feedback throughout the review process has been invaluable in improving the quality and clarity of the article. We have revised the manuscript once again, and all the minor changes made in this round are highlighted in blue.
